# Towards an ensemble-based evaluation of land surface models in light of uncertain forcings and observations

Vivek. K. Arora[1], Christian Seiler[2], Libo Wang[2], and Sian Kou-Giesbrecht[1]

[1]Canadian Centre for Climate Modelling and Analysis, Climate Research Division, Environment Canada, Victoria, BC, Canada

[2]Climate Processes Section, Climate Research Division, Environment and Climate Change Canada, Toronto, ON,
Canada
*Correspondence to*: Vivek K. Arora (vivek.arora@ec.gc.ca)

**Abstract**

Quantification of uncertainty in fluxes of energy, water, and $CO_2$ simulated by land surface models (LSMs) remains a challenge. LSMs are typically driven with, and tuned for, a specified meteorological forcing data set and a specified set of geophysical fields. Here, using two data sets each for meteorological forcing and land cover representation (in which the increase in crop area over the historical period is implemented in the same way), as well as two model structures (with and without coupling of carbon and nitrogen cycles), the uncertainty in simulated results over the historical period is quantified for the Canadian Land Surface Scheme Including Biogeochemical Cycles (CLASSIC) model. The resulting eight (2 x 2 x 2) model simulations are evaluated using an in-house model evaluation framework that uses multiple observations-based data sets for a range of quantities. The simulated area burned, fire $CO_2$ emissions, soil carbon mass, vegetation carbon mass, runoff, heterotrophic respiration, gross primary productivity, and sensible heat flux show the largest spread across the eight simulations relative to their global ensemble mean values. Simulated net atmosphere-land $CO_2$ flux, a critical determinant of the performance of LSMs, is found to be largely independent of the simulated pre-industrial vegetation and soil carbon mass although our framework represents the historical increase in crop area in the same way in both land cover representations. This indicates that models can provide reliable estimates of the strength of the land carbon sink despite some biases in carbon stocks. Results show that evaluating an ensemble of model results against multiple observations disentangles model deficiencies from uncertainties in model inputs, observation-based data, and model configuration.

## 1. Introduction

The current generation land surface models (LSMs) explicitly simulate the fluxes of energy, water, momentum, and trace gases (including $CO_2$, $CH_4$, and $N_2O$) between the atmosphere and the land surface. These models have become an essential tool in understanding what role the land surface plays in the global climate system under current and projected future changes in environmental conditions, including atmospheric $CO_2$ concentration (Bonan and Doney, 2018). LSMs are also an essential component of climate and Earth system models (ESMs), together with their ocean and atmosphere components. Within the framework of ESMs, LSMs are coupled interactively to their atmospheric components through the fluxes of energy, momentum, and matter.

The complexity of LSMs has increased over time as more physical and biogeochemical processes have been included in their framework (Fisher and Koven, 2020; Kyker-Snowman et al., 2022). This increased complexity combined with the uncertainty in our understanding of physical and biogeochemical processes implies that different models respond differently even when driven with the same external forcings. One estimate of the uncertainty in our understanding of land surface physical and biogeochemical processes is obtained by evaluating the inter-model spread in a given quantity when models are forced in the same manner. Other than the uncertainty among models due to differences in their model structures and parameterizations of various processes, uncertainty also exists due to at least three other

reasons. These include uncertainty 1) in parameter values[1] of represented processes, 2) in driving
meteorological data, and 3) in the specification of the geophysical fields. LSMs are typically driven
with meteorological data consisting of seven primary variables (incoming long and shortwave
radiation, temperature, precipitation, specific humidity, wind speed, and pressure).  In addition,
the geophysical fields of land cover, soil texture, and soil permeable depth are also required.
Driving data for LSMs also consist of atmospheric $CO_2$ concentration and other model-specific
external forcings such as nitrogen deposition and fertilizer application rates for models that
include a representation of the terrestrial nitrogen cycle, and lightning, population density, and
gross domestic product (GDP) for models that simulate wildfires.

Every year more than 15 land surface modelling groups participate in the TRENDY (trends

in net land-atmosphere carbon exchanges) project where they perform a set of simulations that
are driven with specified external forcings. The simulations are performed from the year 1700 to
the present day. These simulations contribute to the annual Global Carbon Project's (GCP)
analysis of the land carbon sink together with its analysis of anthropogenic $CO_2$ emissions and
the ocean carbon sink (Friedlingstein et al., 2019). The external forcings used to drive LSMs in the
TRENDY intercomparison include, 1) six hourly meteorological data from 1901 to the present day
(the most recent 2020 TRENDY intercomparison used the CRU-JRA forcing obtained by blending
the climate research unit (CRU) monthly data and the Japanese reanalysis (JRA)); 2) atmospheric
$CO_2$ concentration; and 3) information about changes in crop area and other land use changes
(LUC) from the land use harmonization (LUH) product (Hurtt et al., 2020a). The information about

---

[1] Changes in parameter values do not constitute different parameterizations. For example, two models may use the same parameterization, say $y=mx+b$, but different values of its parameters $m$ and $b$. However, $y=mx + b$ and $y = mx^2$ are considered to be two different parameterizations.

changes in crop area and other LUC is used by land surface modelling groups to reconstruct
historical land cover from the year 1700 to the present day consistent with the number of the
plant functional types (PFTs) a given model represents. The protocol also provides nitrogen
deposition and fertilization application rates for models including nitrogen cycling.

Models participating in the TRENDY simulations are thus driven with common

meteorological and LUC forcings as part of its protocol. The resulting spread across models
participating in the TRENDY project thus provides a measure of inter-model uncertainty, as
mentioned earlier. Traditionally the uncertainty associated with model structure has gained the
most attention and the scientific community has responded to this by performing model
intercomparison projects (MIPs) where models are driven according to a common protocol. The
coupled model intercomparison project (CMIP) in the climate community together with its
various sub-projects (Eyring et al., 2016) is another prominent example. MIPs now routinely form
the basis of evaluating models against observations and multi-model means of various quantities.
Multi-model means are also considered the best estimate for a given quantity (Tebaldi and
Knutti, 2007).

The modelling community has been long aware of the uncertainty associated with

parameter values, since a large fraction of physical and biogeochemical model processes are
parameterized, and such uncertainty analysis dates back to the early hydrological models (e.g.
Hornberger and Spear, 1981; Beven and Binley, 1992). More recent examples of parameter value
uncertainty in the context of a given LSM include Poulter et al. (2010), Booth et al. (2012), and Li
et al. (2018a). The land surface modelling community, however, has only recently begun to
address and quantify uncertainty related with driving meteorological data. Wu et al. (2017), for
example, illustrate the uncertainty in gross primary productivity (GPP) simulated by the Lund-
Potsdam-Jena General Ecosystem Simulator (LPJ-GUESS) model when driven by six different
meteorological data sets. Bonan et al. (2019) analyze the uncertainty in simulated carbon cycle
related variables using three versions of the Community Land Model (CLM) when driven with two
meteorological data sets over the historical period. Slevin et al. (2017) assess the uncertainty in
simulated GPP by the JULES land model when driven by three different meteorological data sets.
Studies that evaluate the effect of different land cover representations on model performance
are even fewer. Tian et al. (2004) and Lawrence and Chase (2007) study the effect of new land
surface boundary conditions, including leaf area index and fractional vegetation cover, based on
the MODIS satellite data as implemented in CLM2 in the Community Atmosphere Model (CAM2)
and CLM3 in the Community Climate System Model (CCSM 3.0), respectively.
Here, we drive the Canadian Land Surface Scheme Including Biogeochemical Cycles
(CLASSIC) with two sets of historical meteorological forcings and also two land cover
representations to quantify the uncertainty associated with both these forcings. Other than
these, we also use two versions of the CLASSIC model: one that represents the interactions
between the carbon (C) and nitrogen (N) cycles and the other in which these interactions are
turned off. CLASSIC has contributed to the simulations for the TRENDY intercomparison, and the
GCP, since 2016 (formerly under the CLASS-CTEM name). Seiler et al. (2021a) have evaluated how
well the CLASSIC model performs when forced with three different meteorological data sets using
the model version without the N cycle. Using the two meteorological forcing data sets, two
representations of land cover, and two versions of the model we perform eight simulations over
the historical period since 1700. All of these simulations may be considered equally likely
representations of the modelled state of the land surface over the historical period. Yet, they all
have their own distinct biases since simulated land surface states and fluxes are different. We
use these simulations to illustrate the uncertainty associated with meteorological forcing and the
two different representations of land cover that are used to drive the model. We also use an in-
house open-source benchmarking system (see code/data availability section) to evaluate these
different simulations against observations-based data sets: AMBER (Automated Benchmarking R
Package) (Seiler et al., 2021b) uses gridded and in-situ observation-based estimates of 19 energy,
water, and C cycle related variables to evaluate LSMs.

Section 2 of this paper describes the framework of the CLASSIC land model and the forcing

data that are required to drive the model. Section 3 describes the two meteorological data sets,
the two representations of land cover that are used to drive the model, and the simulations
performed for this study. Section 4 analyses the results from the simulations to illustrate their
different states and reports results from the AMBER benchmarking exercise. Finally, the
discussion and conclusions are presented in Section 5. The use of more than one meteorological
forcing data sets and land cover representation yields a conundrum since tuning model
parameters for a given forcing data set is not a useful exercise anymore. We also report a new
finding that despite different land C states (characterized in terms of vegetation and soil C mass)
in the eight simulations considered here, the net atmosphere-land $CO_2$ flux over the historical
period in these simulations is consistent with estimates from the GCP. This and the discussion
about the broader question of model tuning are also presented in Section 5.
**2. The CLASSIC land modelling framework**

**2.1 The physical and carbon biogeochemical processes**


The CLASSIC land model is the successor to, and based on, the coupled Canadian Land
Surface Scheme (CLASS; (Verseghy, 1991; Verseghy et al., 1993)) and the Canadian Terrestrial
Ecosystem Model (CTEM; (Arora and Boer, 2005; Melton and Arora, 2016b). CLASSIC also serves
as the land component in the family of Canadian Earth System Models (Arora et al., 2009, 2011;
Swart et al., 2019). Melton et al. (2019) provide an overview of the CLASSIC land model and
launched it as a community model. The basis of the modelling of physical and biogeochemical
processes in CLASSIC comes from CLASS and CTEM, respectively, both of which have a long
history of development. CLASSIC simulates land-atmosphere fluxes of water, energy, and
momentum based on its physics, and fluxes of $CO_2$, $CH_4$, $N_2O$, $NO_x$, and $NH_3$ based on its
biogeochemical process. The representation of the terrestrial N cycle is a new addition to CLASSIC
(Asaadi and Arora, 2021; Kou-Giesbrecht and Arora, 2022) and allows for the simulation of the
interactions between the C and N cycles explicitly.
The CLASSIC model simulations can be performed over a spatial domain, which may be
global or regional, using gridded data or at a point scale, e.g. using meteorological and
geophysical data from a FluxNet site. The primary physical and biogeochemical processes of
CLASSIC are briefly summarized in the next two sections.
**2.1.1 Physical processes**
The calculations for physical processes in CLASSIC are performed over vegetated, snow,
and bare fractions at a time step of 30 minutes. In the version used here, the fractional coverage
of the four plant functional types (PFTs) (needleleaf trees, broadleaf trees, crops, and grasses)
characterizes vegetation for each grid cell. The fractional coverage of these four PFTs is specified
over the historical period in this study. The structure of vegetation is characterized by leaf area
index (LAI), vegetation height, canopy mass, and rooting distribution through the soil layers all of
which are dynamically simulated by the biogeochemical module of CLASSIC. Twenty ground
layers represent the soil profile, starting with 10 layers of 0.1 m thickness. The thickness of layers
gradually increases to 30 m for a total ground depth of over 61 m. The depth of permeable soil
layers and thus the depth to bedrock varies geographically and is specified based on the
SoilGrids250m data set (Hengl et al., 2017). Liquid and frozen soil moisture contents, and soil
temperature, are determined prognostically for permeable soil layers. The temperature, albedo,
mass, and density of a single-layer snow pack (when the climate permits snow to exist) are also
prognostically modelled. The result of physics calculations yields fluxes of energy (primarily net
radiation, ground heat flux, and latent and sensible heat fluxes) and water (primarily
evapotranspiration and runoff) at the land-atmosphere boundary.
**2.1.2 Biogeochemical processes**
The biogeochemical processes in CLASSIC, based on CTEM, are described in detail in the
appendix of Melton and Arora (2016). The biogeochemical processes simulate the land-
atmosphere exchange of $CO_2$ and as a result simulate vegetation as a dynamic component
depending on the environmental conditions.
The biogeochemical module of CLASSIC prognostically calculates the amount of C in the
model's three live (leaves, stem, and root) and two dead (litter and soil) C pools for each PFT. The
live vegetation pools are separated into their structural and non-structural components. The C
amount in these pools is represented per unit land area (kg C/m$^2$). The amount of C in the live
and dead C pools and all terrestrial ecosystem processes in the biogeochemical module in this
study are modelled for nine PFTs that map directly onto the four base PFTs used in the physics
module of CLASSIC. Needleleaf trees are divided into their deciduous and evergreen phenotypes,
broadleaf trees are divided into cold deciduous, drought deciduous, and evergreen phenotypes,
and crops and grasses are divided based on their photosynthetic pathways into C$_3$ and C$_4$
versions. The physical process in CLASSIC are less sensitive to this sub-division of PFTs which is
essential for modelling biogeochemical processes. For instance, simulating the onset and offset
of leaves is different between evergreen and deciduous phenotypes of needleleaf and broadleaf
trees and this is simulated in the biogeochemical module of the model. However, once the leaf
area index (LAI) is known, the physical processes in CLASSIC do not need information about the
underlying deciduous or evergreen nature of leaf phenology. For example, the interception of
rain and snow by canopy leaves (that is typically modelled as a function of LAI and a PFT-
dependent parameter that accounts for leaf orientation and shape) does not depend on the
underlying evergreen or deciduous nature of the leaf phenology. In general, biogeochemical
processes benefit more in terms of realism than physical processes when the number of PFTs is
increased. For example, in offline CLASSIC simulations, large changes in leaf area index (LAI) do
not change total latent heat flux considerably since the partitioning of evapotranspiration into its
sub-components (transpiration, soil evaporation, and evaporation/sublimation of intercepted
rain/snow) adjusts. A decrease in transpiration and evaporation of intercepted precipitation, due
to a decrease in LAI, is compensated by an increase in soil evaporation. This is expected since
water and energy fluxes are determined largely by available energy and precipitation.
The litter and soil C pools are tracked for each soil layer but the movement of C between
the soil layers is not yet modelled. Other than photosynthesis and leaf respiration which are
modelled at a time step of 30 minutes all other biogeochemical processes are modelled at a daily
time step. These include: 1) allocation of C from leaves to stem and roots, 2) autotrophic
respiration from the live C pools and heterotrophic respirations from the dead C pools, 3) leaf
phenology, 4) turnover of live vegetation components that generates litter, 5) mortality, 6) LUC,
and 7) fire (Arora and Melton, 2018). Competition between PFTs for space is not modelled in this
study and fractional coverage of the nine PFTs is specified based on the representation of the
land cover as explained in the next section.
When the N cycle is turned on, land-atmosphere fluxes of $N_2O$, $NO_x$, and $NH_3$, and N
leaching are also modelled in response to biological N fixation, N fertilizer inputs, and N
deposition from the atmosphere. In particular, when the N cycle interacts with the C cycle, the
maximum photosynthetic capacities of model PFTs ($V_{c,max}$) are determined prognostically as a
function of their leaf N content (Asaadi and Arora, 2021; Kou-Giesbrecht and Arora, 2022). When
the N cycle is turned off, prescribed PFT-specific $V_{c,max}$ rates are used (Melton and Arora, 2016a)
and an empirical downregulation parameterization is used to emulate the effect of nutrient
constraints as atmospheric $CO_2$ increases (Arora et al., 2009). N in all model components (leaves,
stem, roots, litter, and soil organic matter) is prognostically tracked, and therefore C:N ratio of
all components is prognostically modelled except for soil organic matter for which a C:N ratio of
13 is specified. In addition, N in the soil mineral pools of nitrate ($NO_3^-$) and ammonium ($NH_4^+$) is
also prognostically modelled.
**3. Driving data for CLASSIC and model simulations**

**3.1 Land cover**

Land cover is one of the most important geophysical fields that is required by LSMs and at its most basic level provides information about fractional vegetation cover in each grid cell for a given regional or global domain. Vegetation in LSMs is typically represented in terms of PFTs. Models may choose to represent a basic set of a few PFTs (trees, grasses, shrubs, and crops) or a more elaborate set that distinguishes PFTs based on their stature (trees, grasses, or shrubs), leaf form (needleleaf or broadleaf), leaf phenology (evergreen or deciduous), photosynthetic pathway ($C_3$ or $C_4$), and geographical location (tropical, temperate, or boreal). The version of CLASSIC in this study uses a somewhat smaller set of nine PFTs for biogeochemical processes as described in the previous section. The fractional coverage of PFTs in a model may be dynamically simulated based on competition between PFTs or prescribed based on observation-based land cover information. While CLASSIC does have a parameterization of competition between its PFTs (Arora and Boer, 2006; Melton and Arora, 2016b), for the historical simulations considered here and for the simulations that contribute to the TRENDY ensemble, prescribed fractional coverage of PFTs is used.

For the process of generating a historical reconstruction of land cover, consisting of time-varying fractional coverage of a model's PFTs, two types of observation-based data sets are used. The first is a remotely-sensed land cover product that represents the geographical distribution of land cover for the present day for a short period. Examples of this include the GLC 2000 land cover product which represents November 1999 to December 2000 period (https://forobs.jrc.ec.europa.eu/products/glc2000/glc2000.php) and the more recent European Space Agency (ESA) Climate Change Initiative (CCI) land cover product for the period 1992-2018

(ESA, 2017). The second type of data set required to reconstruct historical land cover is that of a
spatially and temporally varying cropland (and pasture) area for a much longer period, which in
this case is based on the data set provided by the land use harmonization (LUH) product as part
of the TRENDY protocol for the period 850-2018. The LUH product is comprehensive (Hurtt et al.,
2020b). For example, not all models use the pasture area and other information provided in the
LUH product.
The process of generating land cover for a given model's PFTs is a three-step process.
First, the fractional coverage of model PFTs is obtained from a remotely sensed land cover
product that represents the snapshot of land cover for the present day. This requires typically
mapping 20 – 40 land cover classes that exist in a remotely-sensed land cover product to a given
model's PFTs. This step introduces the largest uncertainty in the entire process. The original land
cover in the CLASSIC model is based on the GLC 2000 land cover product. Table 2 of Wang et al.
(2006) summarizes the mapping/reclassification of the 22 GLC 2000 land cover categories to the
nine PFTs used in CLASSIC. Each land cover class was split into one or more of the nine CLASSIC
PFTs based on the class description and knowledge of global biomes. For example, the discrete
"broadleaf deciduous open tree cover" category of the GLC 2000 product is assumed to consist
of 60% broadleaf deciduous trees, 20% grasses, and 20% bare ground. This first step yields a
snapshot of land cover expressed in terms of the fractional coverage of CLASSIC's nine PFTs
corresponding to the time associated with the land cover product (e.g. for year 2000 for the GLC
2000 land cover product). The second step of generating fractional coverage of PFTs for a given
snapshot in time requires replacing the fractional area of crop categories with values from the
LUH data set for the same year. For example, when using the GLC 2000 land cover product, the
area of $C_3$ and $C_4$ crops from the LUH data set for the year 2000 are used, and the fractional
coverage of the other seven non-crop CLASSIC PFTs is adjusted such that the total vegetation
fraction in each grid cell stays the same. Finally, in the last step, the temporally varying crop area
from the LUH product is used to go backward in time to 1700 from the year 2000 with typically
decreasing crop area while the area of other non-crop PFTs is adjusted in proportion to their
existing fractional coverage such that the total vegetation fraction in each grid cell stays the
same. Similarly, the area of $C_3$ and $C_4$ crops from the LUH product is used from the year 2000
onwards to the present day with changing crop area and the area of non-crop PFTs is adjusted
such that the total vegetation fraction in each grid cell stays the same. All these steps yield a
reconstruction of historical land cover, expressed in terms of fractional coverage of CLASSIC's
nine PFTs (as interpreted from the GLC 2000 land cover product), from 1700 to 2018, in which
crop area changes spatially and temporally according to the LUH product.

GLC 2000 is an older land cover product and more recent land cover products are now

available. Here, in addition to the GLC 2000 based land cover, we also use the European Space
Agency (ESA) Climate Change Initiative (CCI) land cover product. The ESA CCI land cover product
is available at 300 m spatial resolution for the period 1992-2018 and contains 37 land cover
categories (ESA, 2017). We use the land cover from the year 1992 to create a snapshot of CLASSIC
PFTs for the present day. Although there is some interannual variability overall the total
vegetated area doesn't change substantially from 1992-2018 in the ESA-CCI land cover. A default
mapping/reclassification table for converting the ESA CCI classes into PFTs is provided in its user
guide (ESA, 2017). However, it overestimates tree cover along the taiga-tundra transition zone
and underestimates it elsewhere in Canada (Wang et al., 2018, 2019). Wang et al. (2022) have
developed a new reclassification table for converting the 37 ESA CCI land cover categories to
CLASSIC's nine PFTs which is used in this study. A high-resolution land cover map over Canada
and a tree cover fraction data at 30 m resolution are used to compute the sub-pixel fractional
composition of each class in the ESA CCI dataset, which is then used to inform the cross-walking
reclassification procedure (Wang et al., 2022).



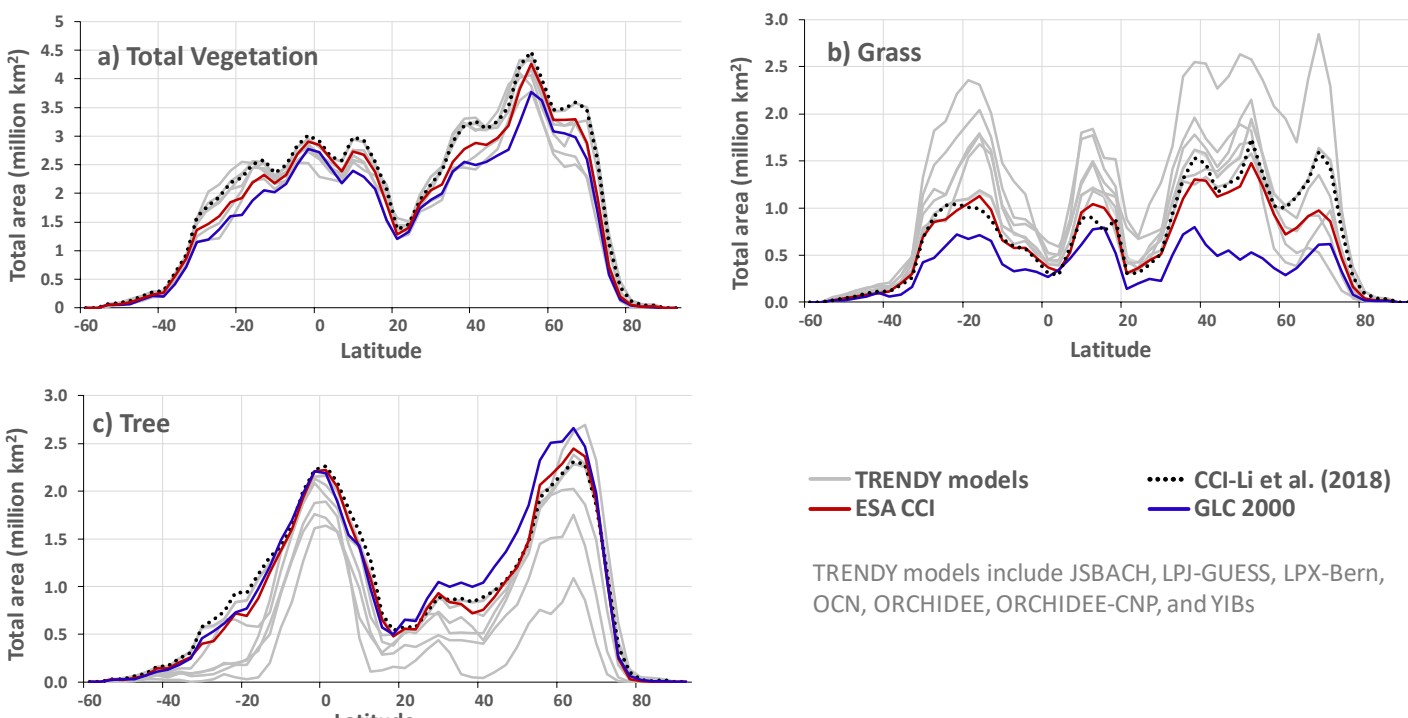


Figure 1: Comparison of zonally summed areas of total vegetation (a), grass (b), and tree (c) cover used
in the CLASSIC model based on GLC 2000 (blue line) and ESA CCI (dark red line) land cover products to
each other, to selected other models that participated in the 2020 TRENDY intercomparison (grey lines)
for which land cover information was available, and to Li et al. (2018) (dotted black line) who analyzed
the ESA CCI data. All data correspond to the 1992-2018 period. CLASSIC does not yet explicitly
represents shrub PFTs. Tall shrubs are merged into tree PFTs in CLASSIC. For the Li et al. (2018) data
plotted here, the shrub PFTs are combined with the tree PFTs for a consistent comparison to CLASSIC.


The above process yields two representations of land cover in which the geographical
distribution of CLASSIC PFTs is based on GLC 2000 and ESA CCI land cover products. Both these
representations include the same reconstruction of crop area over the historical period. Figure 1
illustrates the uncertainty in land cover by comparing zonally summed areas of total vegetation,
tree, and grass cover in CLASSIC, averaged over the period 1992-2018, when model land cover is
based on the GLC 2000 (blue line) and ESA CCI (dark red line) land cover products. These two
estimates are also compared to selected other models that participated in the 2020 TRENDY
intercomparison (grey lines), also for the period 1992-2018, for which land cover information was
available, and to Li et al. (2018b) (dotted black line) who analyzed the ESA CCI data based on the
default reclassification table from the ESA CCI user guide. Figure 1 shows while there is relatively
good agreement across TRENDY models in terms of total vegetation cover there's a much larger
uncertainty in its split between tree and grass PFTs. There are two reasons for the spread in total
vegetated, treed, and grassed areas across TRENDY models. First, modelling groups use different
remotely sensed land cover products for obtaining fractional cover of their model PFTs. Second,
the current process of mapping/reclassifying 20-40 land cover classes of a land cover product to
a model's PFTs is mainly based on the class description and expert judgment which introduces
subjectiveness in the process. Compared to the GLC 2000 based land cover in the CLASSIC model,
the newer ESA CCI based land cover yields a somewhat higher total vegetation cover, a higher
grass cover, and a somewhat lower tree cover area. Unlike the older GLC 2000 based land cover
used in CLASSIC, the newer ESA CCI based grass and tree cover area are within the range of the
TRENDY models reported here. Finally, Figure 1 also allows us to compare the results from the
analysis of Li et al. (2018b) for the ESA CCI land cover (dotted black line) to the ESA CCI
reclassification for CLASSIC (dark red line) by (Wang et al., 2022). Li et al. (2018b) used the default
mapping/reclassification table for converting the ESA CCI classes into PFTs. This comparison
illustrates that the remapping of the ESA CCI land cover classes to CLASSIC's PFTs yields total
vegetation, tree, and grass coverage that is broadly comparable to Li et al. (2018b) although some
differences remain for the grasses.
Our framework accounts for the uncertainty in land cover representation. However, since
both land cover representations in our study account for the increase in crop area over the
historical period in the same way by adjusting the area of non-crop PFTs in proportion to their
existing coverage using the LUH product, our framework is unable to account for the uncertainty
associated with the implementation of LUC. Di Vittorio et al. (2018) quantify this uncertainty by
implementing several approaches to account for the increase in crop area over the historical
period in the framework of an integrated assessment model: by preferentially converting grasses
and shrubs, by preferentially converting forests, and by proportionally adjusting areas of non-
crop PFTs in a way similar to ours. LUC emissions are higher if the increase in crop area is
preferentially obtained by converting forests. A similar uncertainty analysis for LUC emissions is
performed by Peng et al. (2017) using the ORCHIDEE land model who analyze the effect of using
different rules to incorporate the changes in crop and pasture area over the historical period. The
uncertainty related to incorporating LUC information to modify a model's land cover is further
illustrated in Di Vittorio et al. (2014) and Meiyappan and Jain (2012).

**3.2 Meteorological data**
As a land surface component of an ESM, CLASSIC requires meteorological forcing at a sub-
daily temporal resolution. In the offline simulations reported here, the model is run with half-
hourly values of meteorological data (incoming long and shortwave radiation, temperature,
precipitation, specific humidity, wind speed, and pressure). The first meteorological data set used
to drive CLASSIC is from the TRENDY protocol for the year 2020, CRU-JRA v2.1.5, which provides
6 hourly values of the seven variables from the Japanese reanalysis (JRA) with monthly values
adjusted to the climate research unit's data (CRU, https://crudata.uea.ac.uk/cru/data/hrg/). This
yields a blended product from year January 1901 to December 2018 with the 6-hourly temporal
resolution of a reanalysis but without the biases that may be present in reanalysis data (Harris,
2020). The second meteorological data set used here to drive CLASSIC is from the Global Soil
Wetness Project 3 (GSWP3). The GSWP3 forcing data are based on a dynamical downscaling of
the 20[th] century reanalysis (Compo et al., 2011) using a Global Spectral Model (GSM) run at about
50 km resolution. GSM is nudged towards the vertical structures of 20[th] century (20CR) zonal and
meridional air temperature and winds so that the synoptic features are retained at their higher
spatial resolution. Additional bias corrections are also performed as explained in van den Hurk et
al. (2016). The GSWP3 forcing is available for the 1901-2016 period. The 6-hourly values from
both the CRU-JRA and GSWP3 forcings are further disaggregated to half-hourly values for use by
CLASSIC.
Figure A1 compares the two meteorological forcings data sets, over the 1997-2016
period, to illustrate that although these two data sets are very similar there are differences
between the two. Mean annual global precipitation over land (excluding Greenland and
Antarctica) in the GSWP3 data set (71.4 mm/month, 857 mm/year) is somewhat higher than in
the CRU-JRA data set (68.3 mm/month, 820 mm/year). The global near-surface air temperature
over land (excluding Greenland and Antarctica) is also slightly higher in the GSWP3 data set (14.22
°C) compared to the CRU-JRA data set (14.08 °C). The largest temperature difference occurs
between the two data sets over the northern tropics (panel h) where the GSWP3 data set is about
0.93 °C warmer than the CRU-JRA data set. The geographical distribution of mean annual
temperature is very similar between the two data sets but there are some differences in the
geographical distribution of precipitation (not shown). Despite very similar total precipitation
amounts and their seasonality over large global regions in the two data sets, differences exist in
the frequency distribution of precipitation. Figure A2 illustrates this over three broad regions, the
Amazon, the Sahel, and the Midwest United States, which shows the frequency distribution of
daily precipitation amounts (mm/day) over the 1997-2016 period from the two data sets. Figure
A2 shows that the frequency of precipitation events greater than about 5-10 mm/day is higher
in the GSWP3 data set compared to the CRU-JRA data set for the Amazonian, the Sahel, and the
Midwest United States regions.
**3.3 Other forcings**
Other than the land cover and meteorological forcings CLASSIC requires globally averaged
atmospheric $CO_2$ concentration, geographically varying time-invariant soil texture and soil
permeable depth, population density, monthly climatological lightning, and geographically and
time-varying N fertilizer application rates and atmospheric N deposition rates. The atmospheric
$CO_2$ concentration values are provided by the TRENDY protocol. The soil texture information
consists of the percentage of sand, clay, and organic matter and is derived from Shangguan et
al. (2014). N fertilizer is specified according to the TRENDY protocol and based on Lu and Tian
(2017). N deposition is also specified according to the TRENDY protocol and based on model
forcings provided for the sixth phase of CMIP (CMIP6) through input4MIPs (Hegglin et al.,
2016). N deposition for the historical (1850-2014) period is used as is provided while that for
the period 2015-2018 is specified based on N deposition from the SSP5-85 scenario. For the
period 1700-1849, N deposition values from the year 1850 are used.
Table 1: Summary of simulations performed with two representations of the historical land
cover, two sets of meteorological data, and two versions of the CLASSIC land model.

| Simulation | Land cover reconstruction | Meteorological forcing | N cycle interactions with the C cycle |
|---|---|---|---|
| 1 | based on GLC 2000 | CRU-JRA v2.1.5 | On |
| 2 | based on GLC 2000 | GSWP3 | On |
| 3 | based on GLC 2000 | CRU-JRA v2.1.5 | Off |
| 4 | based on GLC 2000 | GSWP3 | Off |
| 5 | based on ESA CCI | CRU-JRA v2.1.5 | On |
| 6 | based on ESA CCI | GSWP3 | On |
| 7 | based on ESA CCI | CRU-JRA v2.1.5 | Off |
| 8 | based on ESA CCI | GSWP3 | Off |


**3.4 Model simulations**
Using the two representations of the historical land cover (based on the GLC 2000 and ESA CCI
land cover products), the two sets of meteorological data (CRU-JRA and GSWP3), and the two
versions of the CLASSIC model (with and without interactions between the C and N cycles) we
perform eight sets of pre-industrial and historical simulations as summarized in Table 1. Pre-
industrial simulations that correspond to the year 1700 are required before doing the historical
simulations (from which we analyze the model results) so that model pools can be spun up to
near equilibrium for each combination of land cover, meteorological forcing, and model
version. The pre-industrial simulations use 1901-1925 meteorological data repeatedly since this
period shows little trends in meteorological variables. Global thresholds of net atmosphere-
land C flux of 0.05 Pg/yr and net atmosphere-land N flux of 0.5 Tg N/yr, in simulations with the
N cycle turned on, are used to ensure the model pools have reached equilibrium. Each historical
simulation is then initialized from its corresponding pre-industrial simulation after it has
reached equilibrium. Simulations driven with the CRU-JRA meteorological data are performed
for the period 1701-2018, and the period 1701-2016 for simulations driven with the GSWP3
meteorological data, although results are reported for the period 1997-2016 which is common
to both simulations. Similar to the pre-industrial simulations, meteorological data from 1901-
1925 is used repeatedly for the period 1701-1900. The global model simulations are performed
at a spatial resolution of about 2.81° (about 312 km at the equator) and the size of the spatial
longitude-latitude grid is $128 \times 64$ grid cells. All model forcings are regridded to this common
spatial resolution. The model is run over about 1900 land grid cells at this resolution excluding
glacial cells in Greenland and Antarctica.

**3.5 Automated benchmarking**
The results from the eight CLASSIC simulations reported here are evaluated using an in-
house model benchmarking system called the Automated Model Benchmarking R package
(AMBER) (Seiler et al., 2021b). AMBER is based on a skill score system originally developed by
(Collier et al., 2018) which is used to quantify model performance and explained in detail in the
appendix. Five scores are used that assess a model's bias ($S_{bias}$), root-mean-square error ($S_{rmse}$),
seasonality ($S_{phase}$), interannual variability ($S_{iav}$), and spatial distribution ($S_{dist}$) against globally
gridded and in-situ data set(s) of observation-based estimates for a given quantity. A score is
computed by first calculating a dimensionless statistical metric, which is then scaled onto a unit
interval, and finally calculating its spatial mean. Scores range from 0 to 1 and are dimensionless.
Higher values indicate better performance. Finally, an overall score $S_{overall}$ is calculated as follows
by giving twice as much weight to $S_{rmse}$
$$S_{overall} = \frac{S_{bias}+2S_{rmse}+S_{phase}+S_{iav}+S_{dist}}{1+2+1+1+1}.\qquad\qquad(1)$$
The decision to give extra weight to $S_{rmse}$ is entirely subjective but follows Collier et al. (2018).

The scores are calculated by comparing gridded and in-situ observation-based estimates,

referred to as reference data sets in Seiler et al. (2021b), for 19 energy (surface albedo, net
shortwave and longwave radiation, total net radiation, latent heat flux, sensible heat flux, ground
heat flux), water (soil moisture, snow, and runoff), and C cycle (GPP, ecosystem respiration, net
ecosystem exchange, net biome productivity, aboveground biomass, soil C, LAI, area burnt, and
fire $CO_2$ emissions) related variables to model simulated quantities. Table 2 summarizes the
source of these observation-based data sets. The resulting model scores express to what extent
simulated and observation-based data agree. A low score does not necessarily indicate poor
model performance. Uncertainties in the meteorological forcing data and geophysical fields used
to drive the model, and/or in the observation-based data itself are possible reasons for the lack
of agreement. One way to assess uncertainties in observation-based data sets is to quantify the
skill score by comparing two independently-derived observation-based data sets (Seiler et al.,
2022). The resulting scores are referred to as benchmark scores and quantify the level of
agreement among the observation-based data sets themselves provided, of course, there are at
Table 2: Observation-based data sets used for model evaluation in AMBER.

| Globally gridded variable(s) | Source | Approach used | Reference |
|---|---|---|---|
| Leaf area index | AVHRR | Artificial neural network | Claverie et al. (2016) |
| Net biome productivity | CAMS | Atmospheric inversion | Agustí-Panareda et al. (2019) |
| Net biome productivity | Carboscope | Atmospheric inversion | Rödenbeck et al. (2018) |
| Surface albedo, net shortwave and longwave radiation, net radiation | CERES | Radiative transfer model | Kato et al. (2013) |
| Net radiation, latent and sensible heat flux, ground heat flux, runoff | CLASSr | Blended product | Hobeichi et al. (2019) |
| Leaf area index | Copernicus | Artificial neural network | Verger et al. (2014) |
| Net biome productivity | CT2019 | Atmospheric inversion | Jacobson et al. (2020) |
| Fire $CO_2$ emissions | CT2019 | Atmospheric inversion | Jacobson et al. (2020) |
| Snow amount | ECCC | Blended product | Mudryk (2020) |
| Liquid soil moisture | ESA | Land surface model | Liu et al. (2011) |
| Area burnt | ESA CCI | Burned area mapping | Chuvieco et al. (2018) |
| Latent and sensible heat flux, gross primary productivity | FLUXCOM | Machine learning | Jung et al. (2019, 2020) |
| Above ground biomass | GEOCARBON | Machine learning | Avitabile et al. (2016); Santoro et al. (2015) |
| Surface albedo, net shortwave and longwave radiation, net radiation | GEWEXSRB | Radiative transfer model | Stackhouse et al. (2011) |
| Area burnt | GFED 4s | Burned area mapping | Giglio et al. (2010) |
| Gross primary productivity | GOSIF | Statistical model | Li and Xiao (2019) |
| Soil carbon | HWSD | Soil inventory | Wieder (2014); Todd-Brown et al. (2013) |
| Surface albedo | MODIS | Bidirectional Reflectance Distribution Function | Strahler et al. (1999) |
| Gross primary productivity | MODIS | Light use efficiency model | Zhang et al. (2017) |
| Leaf area index | MODIS | Radiative transfer model | Myneni et al. (2002) |
| Soil carbon | SGS250m | Machine learning | Hengl et al. (2017) |
| Above ground biomass | Zhang | Data fusion | Zhang and Liang (2020) |
| | | | |
| In situ variable(s) | Source | Approach used (number of sites) | Reference |
| Leaf area index | CEOS | Transfer function (141) | Garrigues et al. (2008) |
| Latent, sensible, and ground heat flux, gross primary productivity, ecosystem respiration, net ecosystem exchange | FLUXNET 2015 | Eddy covariance (204) | Pastorello et al. (2020) |
| Above ground biomass | FOS | Allometry (274) | Schepaschenko et al. (2019) |
| Runoff | GRDC | Gauge records (50) | Dai and Trenberth (2002) |
| Snow amount | Mortimer | Gravimetry (3271) | Mortimer et al. (2020) |
| Above ground biomass | Xue | Allometry (1974) | Xue et al. (2017) |


least two sets of observation-based data for a given quantity. The comparison of model scores
against benchmark scores then shows how well a model-simulated quantity compares to the
reference data sets relative to the agreement between the observation-based data sets
themselves.

**4. Results**

Figures 2 through 9 show the time series and/or zonally-averaged values of annual values

of a variable of interest when averaged across four ensemble members each according to
whether the N cycle is turned on or not, whether the GLC 2000 or ESA CCI based land cover is
used, and whether model simulations are driven by the CRU-JRA or GSWP3 meteorological data.
Figures A3, A4, A6, A7, A9, and A11 in the appendix, which are complementary to the above-
mentioned figures, show the physical and biogeochemical states of the land surface and primary
physical fluxes of water and energy, and primary biogeochemical fluxes of $CO_2$ simulated by
CLASSIC at the land-atmosphere boundary for all the eight simulations considered here. While
the figures in the appendix illustrate the range in simulated physical and biogeochemical states
and fluxes across the eight simulations, Figures 2 through 9 evaluate the effect of model
structure, meteorological forcing, and land cover on a given quantity. We also quantify the spread
across the eight simulations in Table 3 using the coefficient of variation (cv= standard
deviation/mean) calculated using annual global values for a given quantity averaged over the
1997-2016 20-year period of each simulation. This time period is also used for other reported
results.
**4.1 Physical land surface state and fluxes**

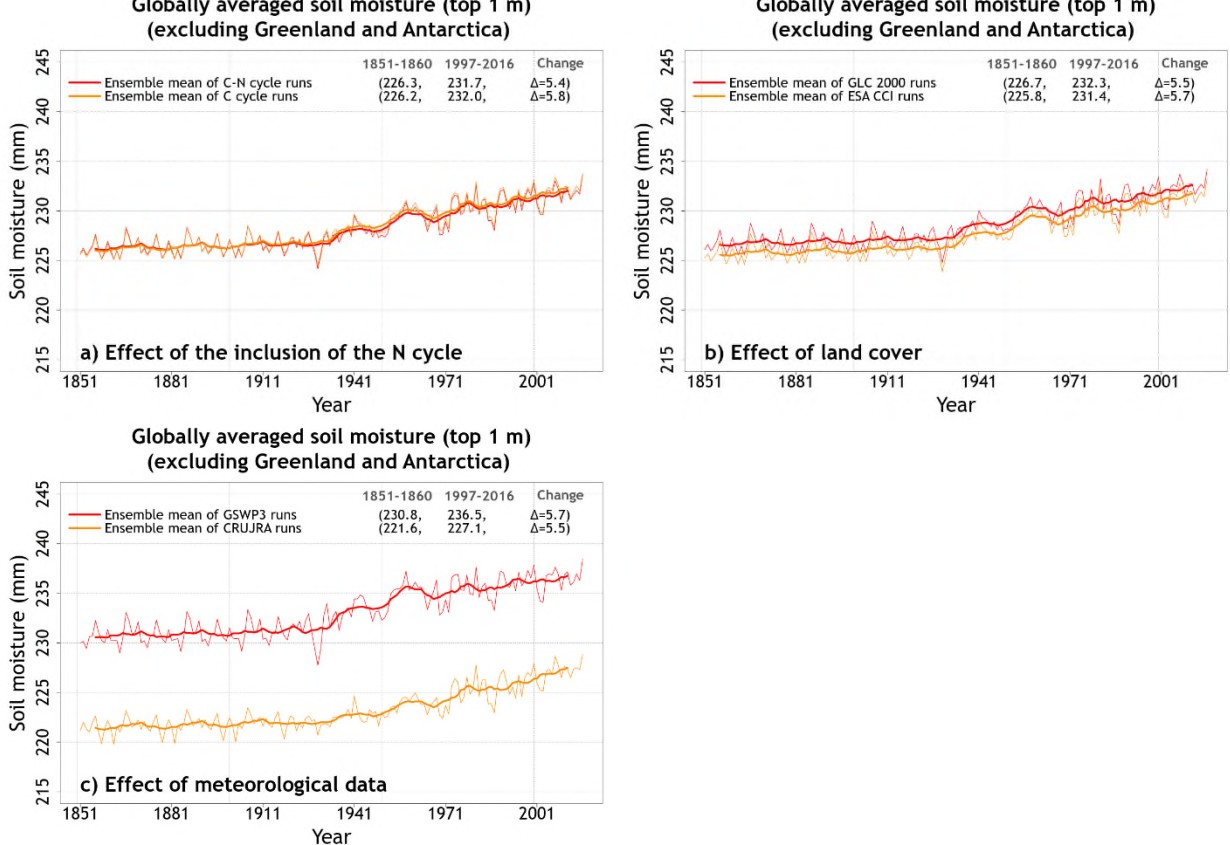

Figure 2: Time series of annual globally-averaged soil moisture in the top 1m averaged over the four ensemble members that are driven with and without an interactive N cycle (panel a), driven with the GLC 2000 and ESA CCI based land cover representations (panel b), and driven with the GSWP3 and CRU-JRA meteorological data (panel c). The thin lines show the individual years and the thick lines show their 11-year moving average. Model values averaged over the pre-industrial (1851-1860) and present-day (1997-2016) time periods, and their difference, for each ensemble averaged over its set of four simulations are also shown.

Figure A3, panels a and b, shows the globally-averaged simulated soil moisture and temperature in the top 1 m soil layer. While simulated soil temperature in the top 1 m is fairly similar across the eight simulations, the simulated soil moisture is distinctly separated into two groups. The separation into these two groups is caused by the driving meteorological data as shown in Figure 2. The coefficient of variation for soil moisture and temperature values averaged over the 1997-2016 period of each simulation are 0.02 and 0.004, respectively, indicating that overall the variation in these quantities is relatively small compared to their means. The use of

the GSWP3 meteorological dataset yields slightly higher (~4%) globally-averaged soil moisture
compared to the CRU-JRA meteorological data set (236.5 mm vs. 227.1 mm, Figure 2c) for the
1997-2016 period.

Figure A3, panels c and d, shows the simulated fluxes of global evapotranspiration and

runoff across the eight simulations. Similar to soil moisture, evapotranspiration and runoff also
fall broadly into two groups and the reason for this again is the driving meteorological data.
Figure 3 shows that while the biggest factor that affects evapotranspiration and runoff is the
difference in driving meteorological data the interactive N cycle also affects these water fluxes.
Neither evapotranspiration nor runoff is significantly affected by the choice of land cover. The
reason an interactive N cycle affects evapotranspiration is that the N cycle in CLASSIC affects the
rate of photosynthesis through the prognostic determination of leaf N content. Photosynthesis
in turn affects canopy conductance, which affects transpiration through the canopy leaves.
Average evapotranspiration over the 1997-2016 period of the simulations driven with GSWP3
meteorological data is about 9% lower than in simulations driven with CRU-JRA meteorological
data (65.8 vs. 72.1 ×1000 km$^3$/year, Figure 3, panel e). An interactive N cycle reduces
evapotranspiration by about 2% due to lower photosynthesis rates as shown later (Figure 3, panel
a). Average runoff is about 27% higher in simulations driven with GSWP3 compared to
simulations driven with CRU-JRA meteorological data (52.6 vs 41.3 ×1000 km$^3$/year, Figure 3,
panel f). This is due to slightly high precipitation in the GSWP3 meteorological data set (Figure
A1) but is more so due to the simulated lower evapotranspiration when using the GSWP3 data
(Figure 3, panel e). The coefficient of variation for evapotranspiration and runoff values averaged
over the last 20 years of each simulation are 0.05 and 0.13, respectively.

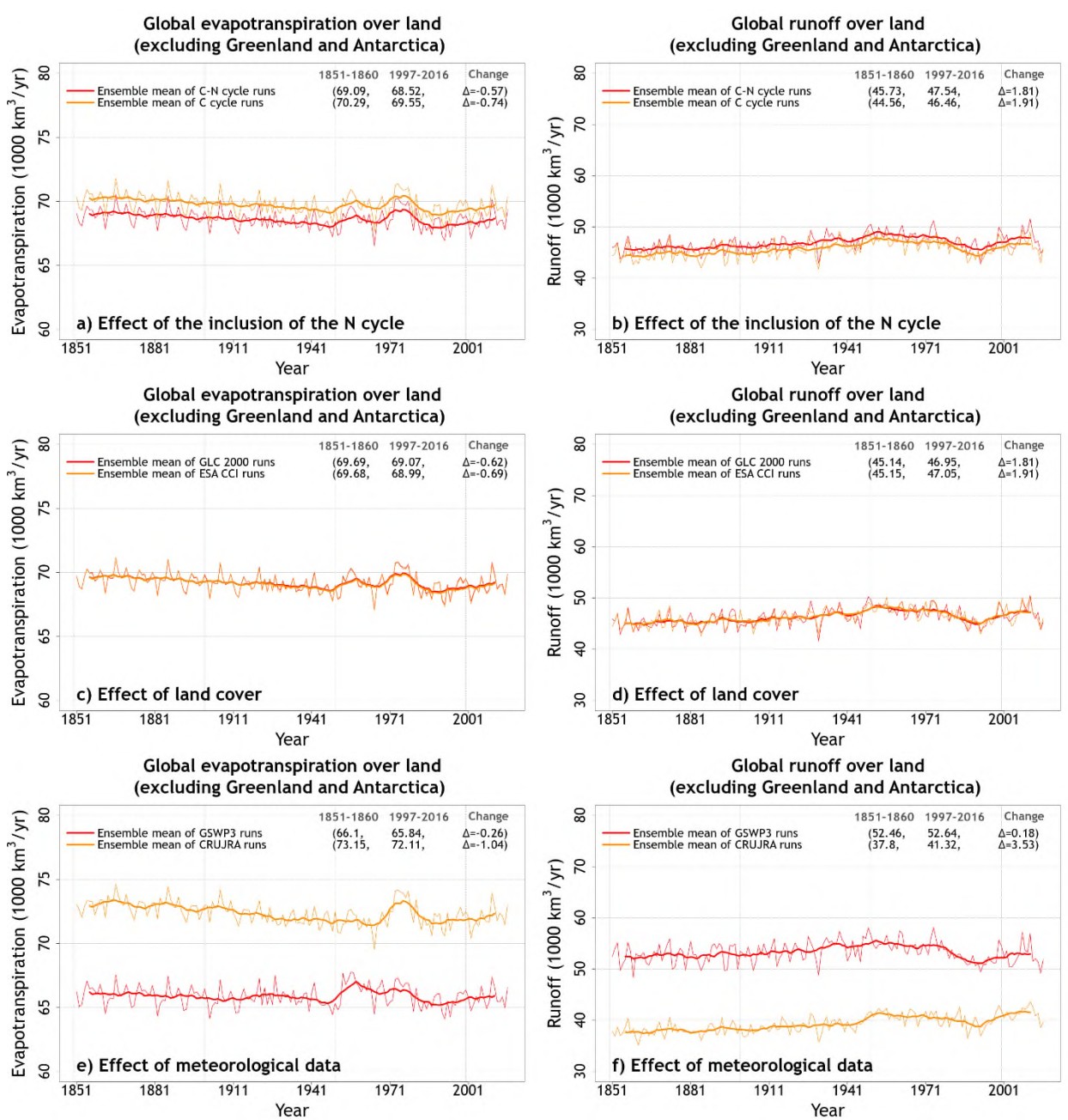


Figure 3: Time series of annual global evapotranspiration and runoff (over all land area excluding Greenland and Antarctica) averaged over the four ensemble members that are driven with and without an interactive N cycle (panels a, b), driven with the GLC 2000 and ESA CCI based land cover (panels c, d), and driven with the GSWP3 and CRU-JRA meteorological data (panels e, f). The thin lines show the individual years and the thick lines show their 11-year moving average. Model values averaged over the pre-industrial (1851-1860) and present-day (1997-2016) time periods, and their difference, for each ensemble averaged over its set of four simulations are also shown.

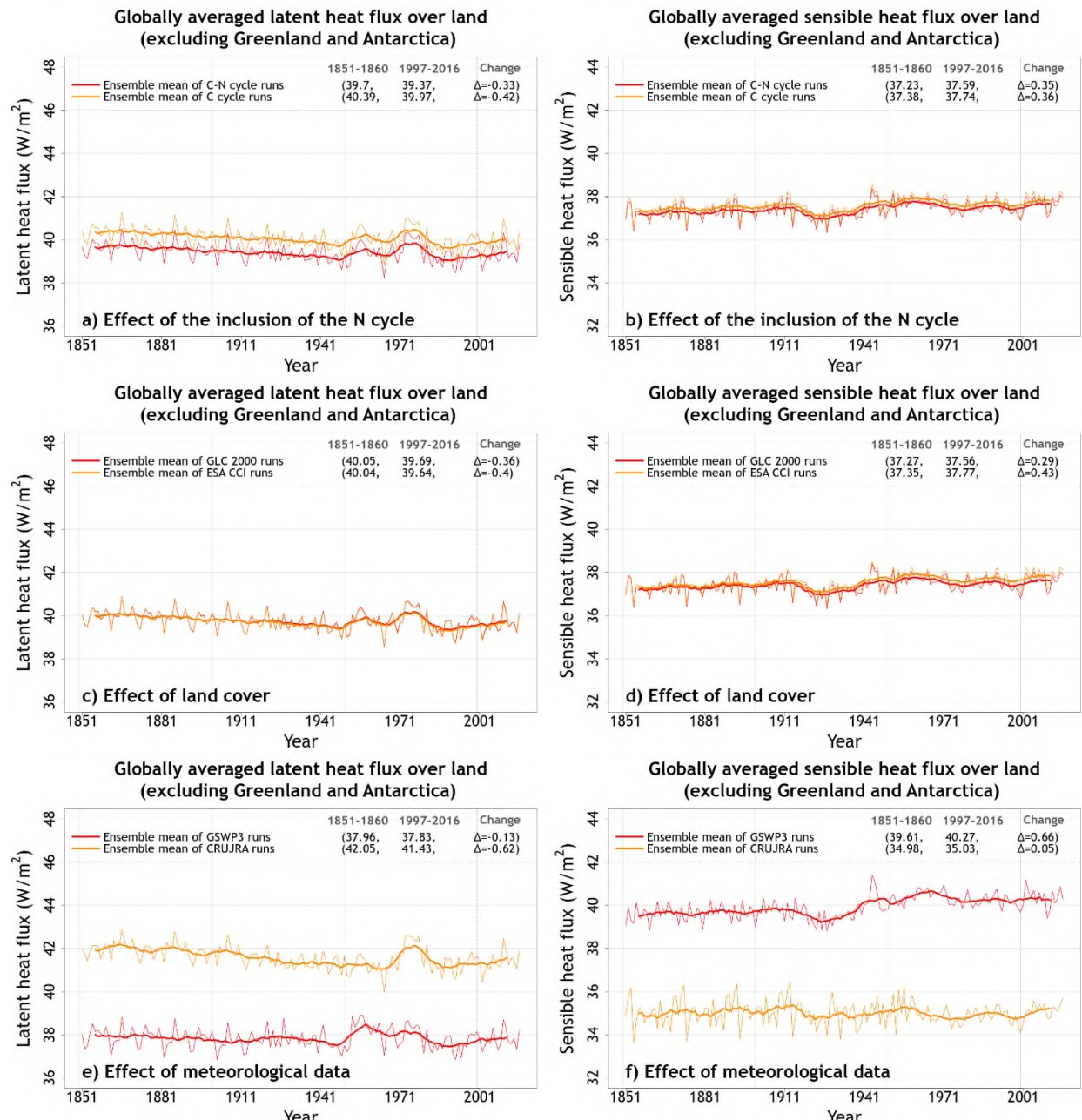

Figure 4: Time series of annual global latent and sensible heat fluxes (over all land area excluding Greenland and Antarctica) averaged over the four ensemble members that are driven with and without an interactive N cycle (panels a, b), driven with the GLC 2000 and ESA CCI based land cover (panels c, d), and driven with GSWP3 and CRU-JRA meteorological data (panels e, f). The thin lines show the individual years and the thick lines show their 11-year moving average. Model values averaged over the pre-industrial (1851-1860) and present-day (1997-2016) time periods, and their difference, for each ensemble averaged over its set of four simulations are also shown.

Figure A4 shows the primary energy fluxes from the eight simulations. These include net

downward shortwave and longwave radiation, and latent and sensible heat fluxes. Incoming
shortwave and longwave radiation are part of the driving meteorological data. Similar to water
fluxes, the differences in energy fluxes in CLASSIC are also primarily driven by differences in
meteorological data (Figure A4, A5, and Figure 4). Net shortwave radiation (Figure A4, panel a) is
equal to incoming shortwave radiation minus the fraction that is reflected back. Net longwave
radiation (Figure A4, panel b) is equal to incoming longwave radiation minus the longwave
radiation emitted by the land based on its surface temperature following the Stefan-Boltzmann
law. The difference in net shortwave radiation is also affected by simulated vegetation biomass
and leaf area index. The latter affects surface albedo which determines what fraction of incoming
shortwave radiation is reflected. This is the reason why an interactive N cycle affects net
shortwave radiation since the N cycle affects photosynthesis, and in turn, simulated vegetation
biomass and leaf area index (Figure A5, panel b). Latent heat flux is affected primarily by
meteorological data (Figure 4) but also if the N cycle is interactive or not since it is essentially
evapotranspiration but in energy units. Finally, differences in sensible heat fluxes are strongly
affected by differences in driving meteorological data (Figure 4). Globally-averaged sensible heat
flux in the simulations driven with GSWP3 data is ~14% higher compared to CRU-JRA driven
simulations (40.2 vs. 35.0 W/m$^2$). The coefficient of variation for latent and sensible heat flux
values averaged over the last 20 years of each simulation are 0.05 and  0.07, respectively. Net
shortwave (cv=0.006) and longwave (cv=0.03) radiative fluxes vary little across the eight
simulations.

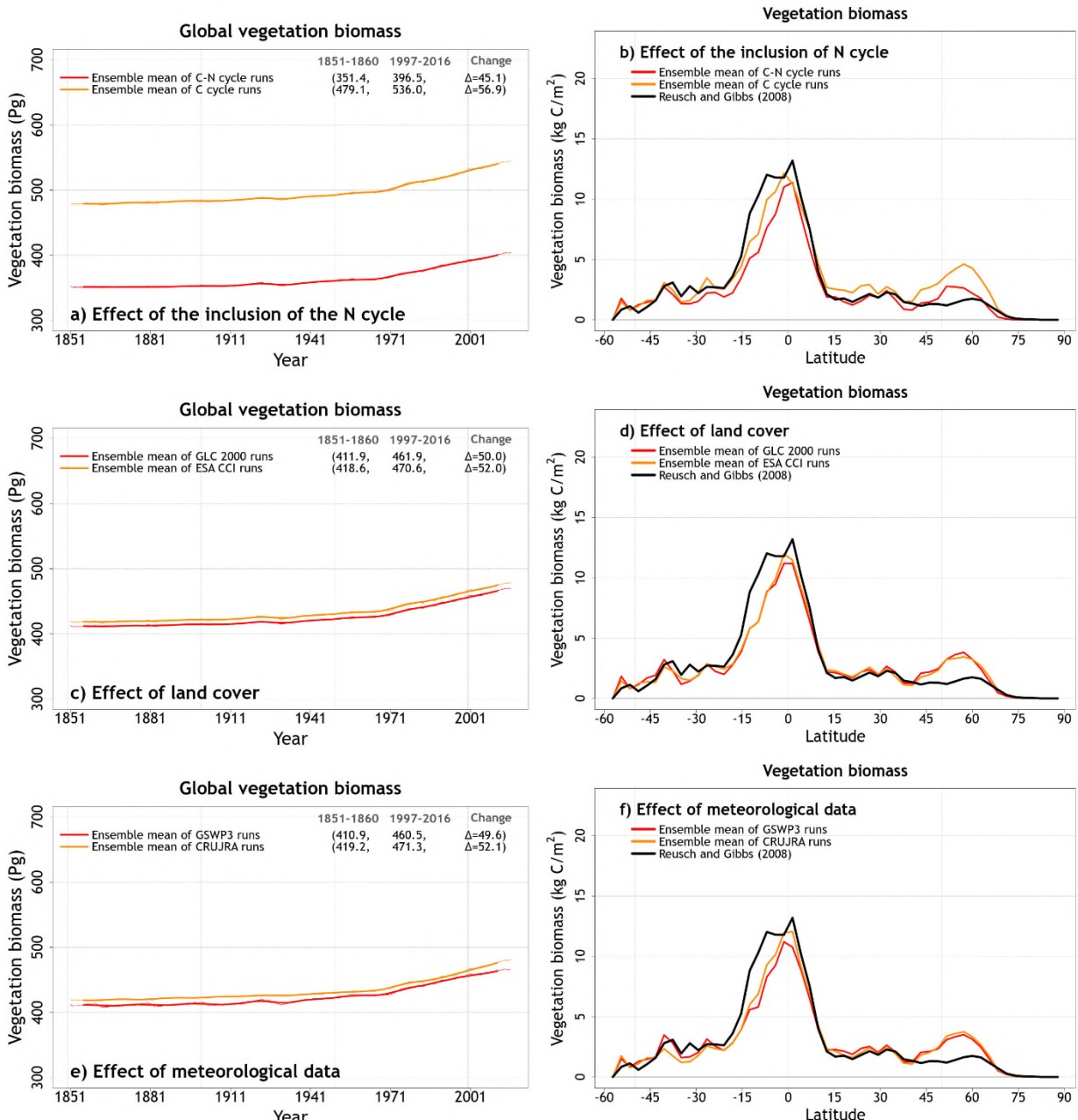

Figure 5: Time series of annual global vegetation C mass (over all land area excluding Greenland and Antarctica) (panels a, c, and e) and zonally-averaged values of vegetation C mass over land (panels b, d, and f) averaged over the four ensemble members, for the period 1997-2016, that are driven with and without an interactive N cycle (panels a, b), driven with the GLC 2000 and ESA CCI based land cover (panels c, d), and driven with GSWP3 and CRU-JRA meteorological data (panels e, f). The thin lines for the time series show the individual years and the thick lines show their 11-year moving average. Model values averaged over the pre-industrial (1851-1860) and present-day (1997-2016) time periods, and their difference, are also shown in panels a, c, and e.


**4.2 Biogeochemical land surface state and fluxes**

**4.2.1 Primary CO$_2$ fluxes and C pools**

Figure A6 shows the simulated C state of the land surface expressed in terms of vegetation
and soil C pools. Panels a and b show the annual time series of global vegetation and soil C mass
from the eight simulations, and panels c and d show their zonally-averaged distributions
averaged over the last 20 years of each simulation. The biggest difference in the time series of
global vegetation (cv=0.16) and soil (cv=0.21) C mass compared to soil moisture and
temperature, which characterized the physical land surface state, is the large spread across the
eight simulations as indicated by their high cv values. The zonally-averaged values further provide
insight into the reasons for this spread and show that the largest differences between simulated
vegetation and soil C occur at northern high latitudes (north of about 40°N). Panels c and d of
Figure A6 also show observation-based zonally-averaged values of vegetation and soil C mass
based on the Reusch and Gibbs (2008) and the Harmonized World Soils Database (v1.2) (Fischer
et al., 2008), respectively, to provide a reference. A more thorough comparison with observations
is provided in Section 4.3.
Differences in vegetation C mass are caused primarily when the N cycle is interactive or
not (Figure 5). Both land cover and the driving meteorological data play a smaller role in the
simulated spread of vegetation C mass (Figure 5). The ESA CCI based land cover has a larger
vegetated area but most of this increase comes from an increase in the area of grasses that do
not store a lot of C in their vegetation C mass. The spread in simulated soil C is caused due to the

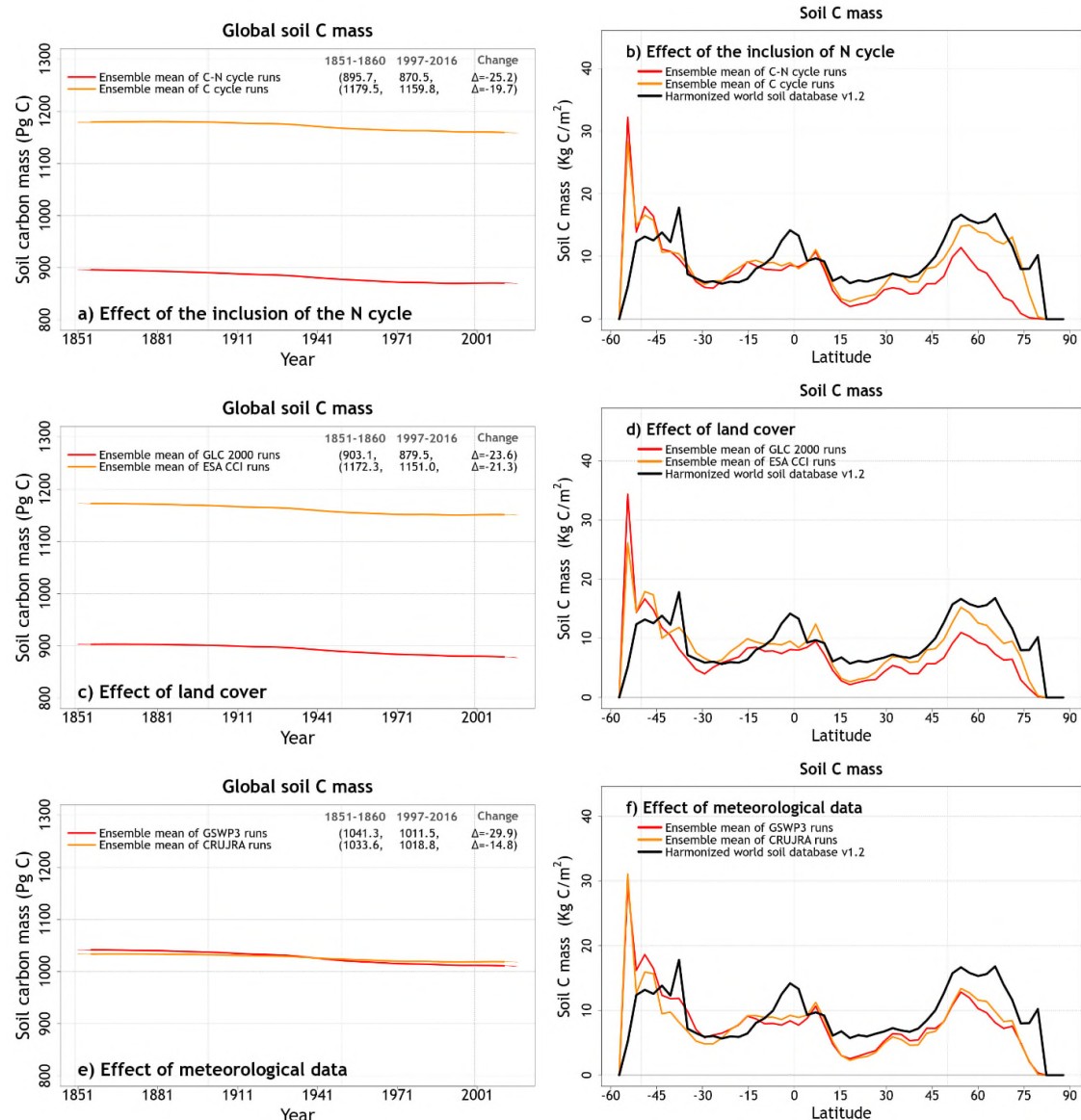



Figure 6: Time series of annual global soil carbon mass (over all land area excluding Greenland and Antarctica) (panels a, c, and e) and zonally-averaged values of soil carbon mass over land (panels b, d, and f) averaged over the four ensemble members, for the period 1997-2016, that are driven with and without an interactive N cycle (panels a, b), driven with the GLC 2000 and ESA CCI based land cover (panels c, d), and driven with GSWP3 and CRU-JRA meteorological data (panels e, f). The thin lines for the time series show the individual years and the thick lines show their 11-year moving average. Model values averaged over the pre-industrial (1851-1860) and present-day (1997-2016) time periods, and their difference, are also shown in panels a, c, and e.

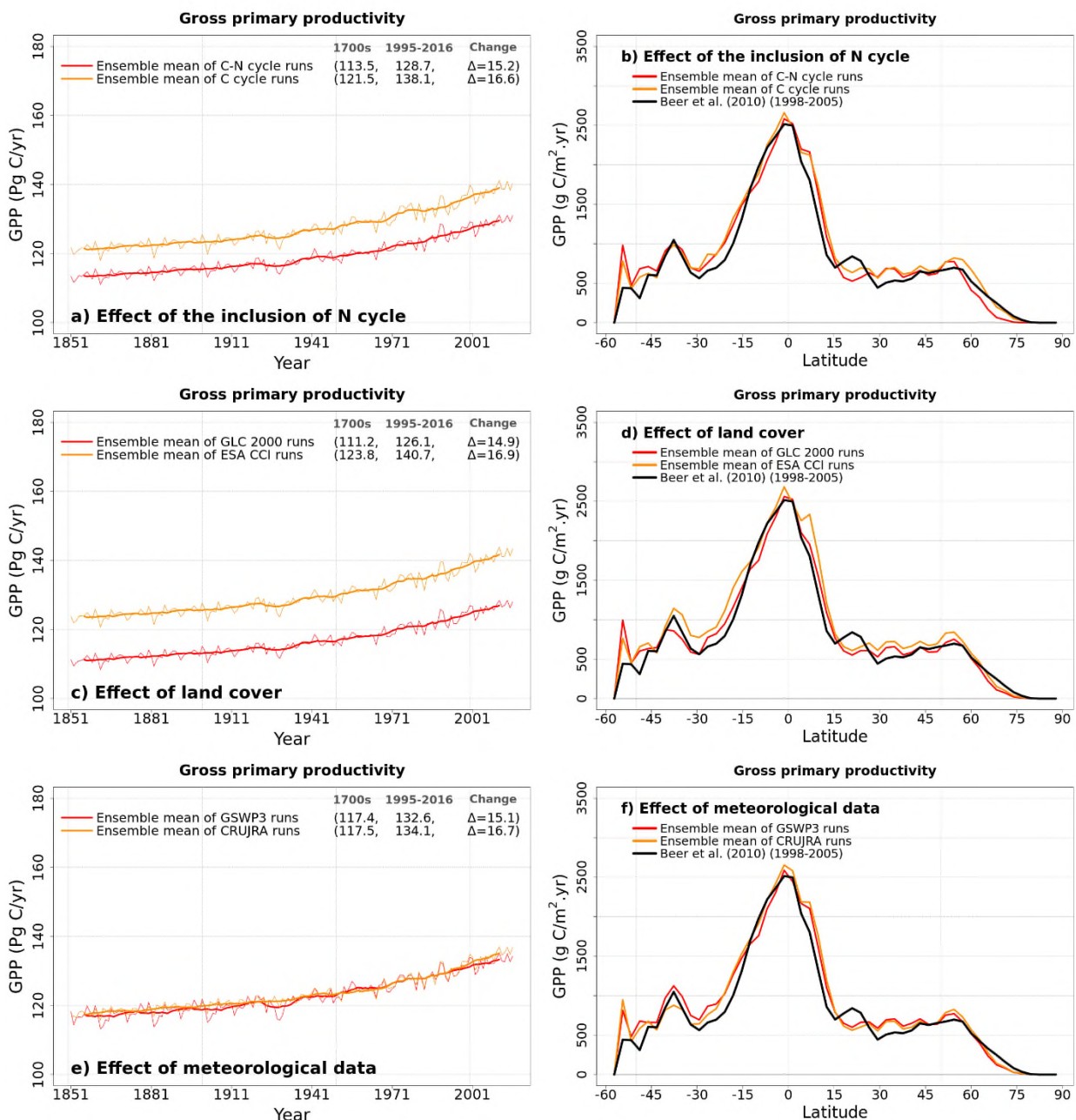

Figure 7: Time series of annual global gross primary productivity (over all land area excluding Greenland and Antarctica) (panels a, c, and e) and zonally-averaged values of gross primary productivity over land (panels b, d, and f) averaged over the four ensemble members, for the period 1997-2016, that are driven with and without an interactive N cycle (panels a, b), driven with the GLC 2000 and ESA CCI based land cover (panels c, d), and driven with GSWP3 and CRU-JRA meteorological data (panels e, f). The thin lines for the time series show the individual years and the thick lines show their 11-year moving average. Model values averaged over the pre-industrial (1851-1860) and present-day (1997-2016) time periods, and their difference, are also shown in panels a, c, and e.

N cycle but also the choice of land cover (Figure 6). Since CLASSIC assumes that litter from grasses
is more recalcitrant than that from trees, the choice of ESA CCI based land cover leads to a higher
soil C mass because it has a higher grass area than the GLC 2000 based land cover (Figure 6,
panels c and d). The choice of meteorological data does not affect the magnitude of simulated
globally-summed soil C mass significantly but does affect its change over the historical period. In
Figure 6 (panel c) the decrease in soil C mass from the 1851-1860 period to the 1997-2016 period
is higher when using the GSWP3 (29.9 Pg C) compared to when using the CRU-JRA (14.8 Pg C)
meteorological data.

The reason why an interactive N cycle in CLASSIC affects vegetation C and soil C mass, and

why the ESA CCI based land cover yields high soil C, is seen in Figures A7 and 7. Figure A7 shows
the spread of primary C fluxes including gross primary productivity (GPP) (cv=0.07), and
autotrophic (cv=0.04) and heterotrophic (cv=0.10) respiratory fluxes, across the eight
simulations.  Since GPP is lower in the runs with the N cycle, both vegetation (Figure 5a) and soil
C mass (Figure 6a) are also lower. The lower GPP in the runs with the N cycle is due primarily to
lower GPP at high latitudes (Figure 7b) which yields low vegetation C mass at high latitudes
(Figure 5b). Low GPP at high latitudes translates to even larger relative differences in soil C given
the longer turnover time scales of soil C at high latitudes (Figure 6b). The use of the ESA CCI based
land cover which has a higher grass area than the GLC 2000 based land cover leads to higher GPP
(Figure 7d) and therefore higher soil C at all latitudes (Figure 6d). In Figure A8, global
heterotrophic and autotrophic respiratory fluxes are most affected by land cover and the
inclusion or absence of an interactive N cycle but not as much by the driving meteorological data.

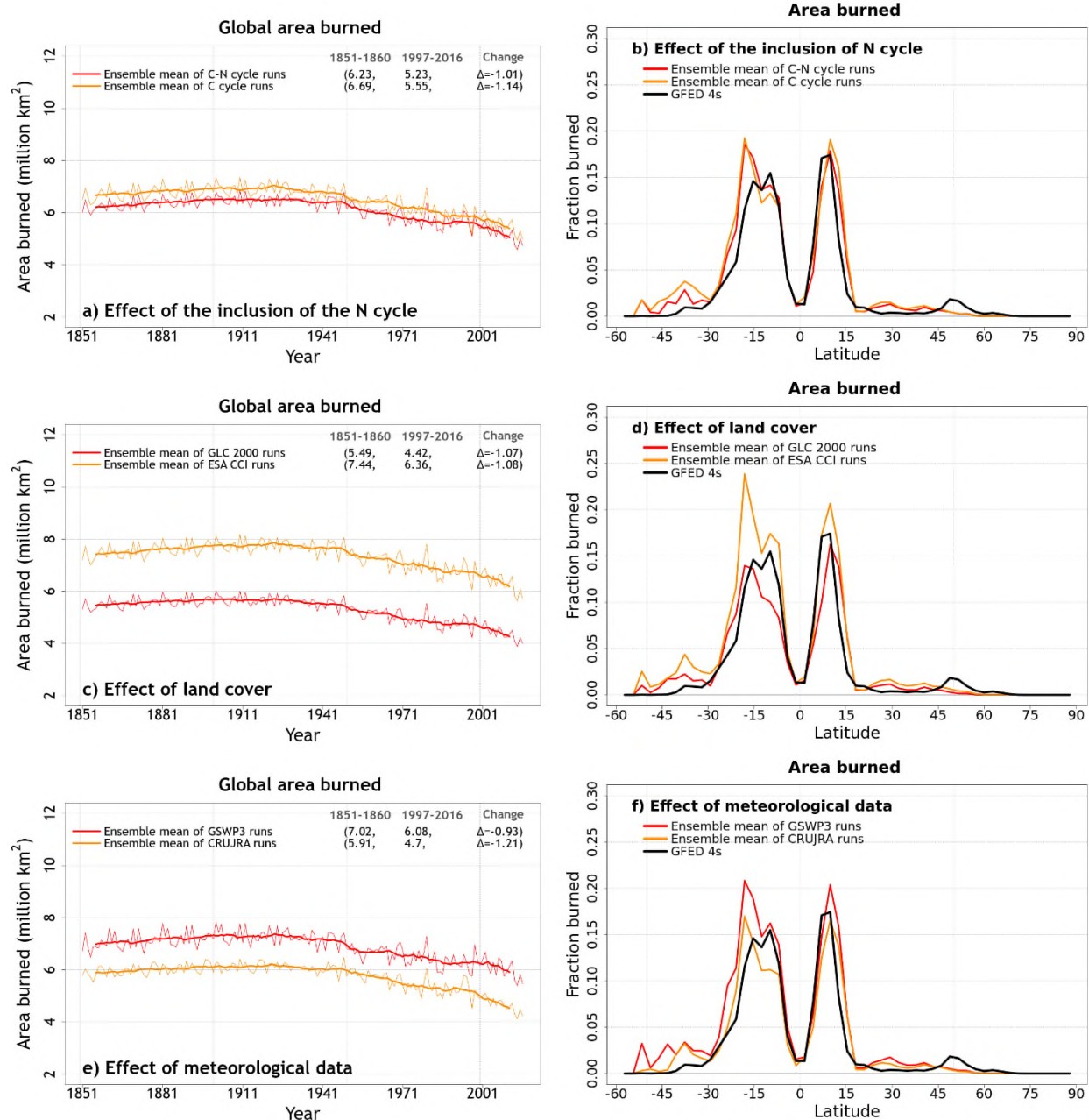


Figure 8: Time series of annual area burned (over all land area excluding Greenland and
Antarctica) (panels a, c, and e) and zonally-averaged values of area burned (panels b, d, and f)
averaged over the four ensemble members, for the period 1997-2016, that are driven with and
without an interactive N cycle (panels a, b), driven with the GLC 2000 and ESA CCI based land
cover (panels c, d), and driven with GSWP3 and CRU-JRA meteorological data (panels e, f). The
thin lines for the time series show the individual years and the thick lines show their 11-year
moving average in panels (a), (c), and (e). Model values averaged over the pre-industrial (1851-
1860) and present-day (1997-2016) time periods, and their difference, are also shown for
panels (a), (c), and (e).


The transient behaviour of heterotrophic respiration over the historical period is not affected by
meteorological data, although the effect of meteorological data on autrotrophic respiration
varies over time.
**4.2.2 Area burned and fire $CO_2$ emissions**

Figure A9 shows the time series of global area burned and global fire $CO_2$ emissions, and

their zonally-averaged values. We chose the area burned (cv=0.24) and fire $CO_2$ emissions
(cv=0.21) in addition to the primary biogeochemical fluxes since fire shows large variability both
in space and in time, and both these variables yield the largest spread across the eight
simulations, among all the fluxes and simulated quantities considered here. Figures A9 (panels c
and d) also show observation-based estimates for area burned and fire $CO_2$ emissions based on
GFED 4s (Giglio et al., 2013) to provide an observation-based context. Figures 8 and A10 help us
understand which factors contribute to this large variability. The variability in the area burned is
caused primarily by the choice of land cover and meteorological data and the variability is higher
in the southern hemisphere (Figure 8, panels d and f). An interactive N cycle does not affect the
zonal distribution of area burned and fire $CO_2$ emissions (Figures 8 and A10) as much. The reason
both area burned and fire $CO_2$ emissions are affected by the choice of land cover is because the
ESA CCI land cover has higher grass area and, as a result, it yields higher area burned and fire $CO_2$
emissions since a larger area is burned for grasses than for trees in the model. The choice of
driving meteorological data is a factor in the area burned and our simulations show that the use
of GSWP3 meteorological forcing yields a higher area burned than the CRU-JRA data. In particular
wind speed, which determines the rate of spread of fire in CLASSIC, is much higher in the GWSP3
than in the CRU-JRA meteorological data. Globally-averaged land wind speed (excluding
Greenland and Antarctica) in GSWP3 data is 6.1 m/s compared to 3.4 m/s in the CRU-JRA data
for the period 2000-2016.
Table 3: Simulated energy, water, and carbon cycle quantities considered in this study sorted
according to their coefficient of variation. The quantities are listed from the most variable at
the top to the least variable at the bottom. The coefficient of variation is based on annual
values averaged over the 1997-2016 period across the eight simulations. The last column shows
the dominant source of variability for each model simulated quantity.

| Energy, water, or carbon cycle quantities | Coefficient of variation | Dominant source of variability |
|---|---|---|
| Area burned (million km$^2$) | 0.24 | Land cover |
| Fire $CO_2$ emissions (Pg C/year) | 0.21 | Land cover |
| Soil carbon mass (Pg C) | 0.21 | The inclusion or the absence of the N cycle |
| Vegetation carbon mass (Pg C) | 0.16 | The inclusion or the absence of the N cycle |
| Runoff (1000 km$^3$/year) | 0.13 | Meteorological forcing |
| Leaf area index (m$^2$/m$^2$) | 0.11 | The inclusion or the absence of the N cycle |
| Heterotrophic respiration (Pg C/year) | 0.10 | Land cover |
| Gross primary productivity (Pg C/year) | 0.07 | Land cover |
| Sensible heat flux (W/m$^2$) | 0.07 | Meteorological forcing |
| Autotrophic respiration (Pg C/year) | 0.04 | Land cover |
| Latent heat flux (W/m$^2$) / Evapotranspiration (1000 km$^3$/year) | 0.05 | Meteorological forcing |
| Net longwave radiation (W/m$^2$) | 0.03 | Meteorological forcing |
| Soil moisture in the top 1m soil layer (mm) | 0.02 | Meteorological forcing |
| Albedo for shortwave radiation (fraction) | 0.008 | The inclusion or the absence of the N cycle |
| Net shortwave radiation (W/m$^2$) | 0.006 | Meteorological forcing |
| Soil temperature in the top 1m soil layer ($^\circ$C) | 0.004 | Meteorological forcing |


### 4.2.3 Coefficient of variation summary

Table 3 shows the energy, water, and C-related quantities considered so far but also leaf
area index and albedo and lists them from the most variable at the top to the least variable at
the bottom according to their coefficient of variation. The area burned is found to be the most
variable quantity and soil temperature is the least variable quantity. Table 3 also shows the most
dominant source of variability for each simulated quantity: land cover, meteorological forcings,
or the inclusion or absence of an interactive N cycle.  Net atmosphere-land $CO_2$ flux (or net biome
productivity), net ecosystem exchange, and ground heat flux are not included in Table 3 because
these fluxes are calculated as the difference of larger fluxes and as a result, their values are closer
to zero which yields a large value of the coefficient of variation. Net surface radiation is the sum
of net shortwave and longwave radiation and both of them exhibit low coefficient of variability
across the eight simulations (Table 3).
**4.2.4 Model tuning**
Overall, the results presented so far illustrate that different model simulated quantities
are sensitive to different forcings and model versions. The use of more than one meteorological
forcing data sets and land cover representation, and the use of two model versions (with and
without N cycle), yields a dilemma since it is no longer possible to tune model parameters without
choosing a preferred meteorological data set, land cover representation, and model version. As
such it seems logical that rather than tuning the model for a preferred forcing or model version,
model results from an ensemble of simulations be compared against an ensemble of
observations in so long as it is possible. This is the approach taken in Section 4.3 with automated
benchmarking.
**4.2.5 Net biome productivity**

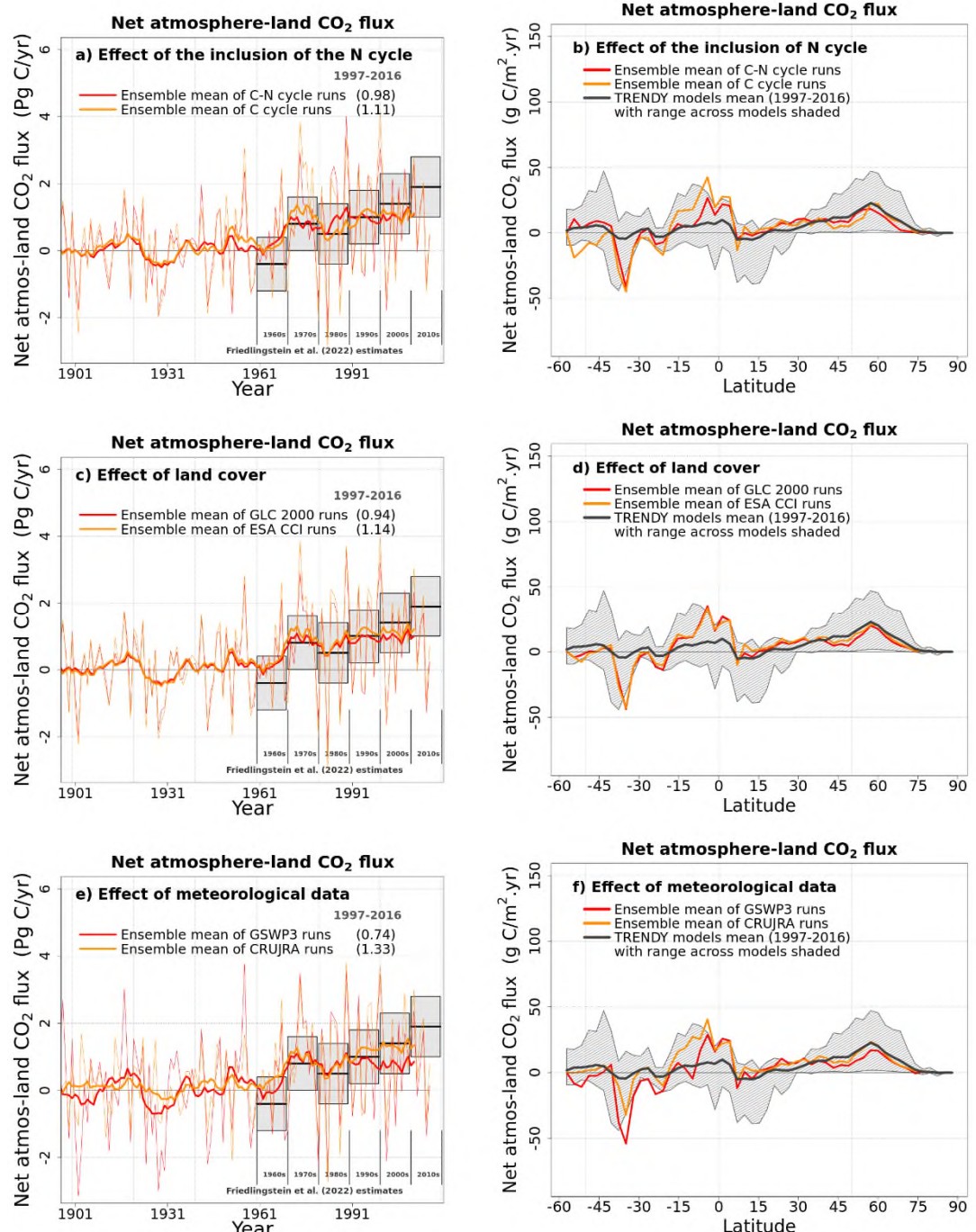

Figure 9: Time series of global net atmosphere-land CO₂ flux (over all land area excluding Greenland and Antarctica) (panels a, c, and e) and its zonally-averaged values (panels b, d, and f) averaged over the four ensemble members, for the period 1997-2016, that are driven with and without an interactive N cycle (panels a, b), driven with the GLC 2000 and ESA CCI based land cover (panels c, d), and driven with GSWP3 and CRU-JRA meteorological data (panels e, f). The thin lines for the time series show the individual years and the thick lines show their 11-year moving average. Model values averaged over the pre-industrial (1851-1860) and present-day (1997-2016) time periods, and their difference, are also shown for panels (a), (c), and (e).

Figure A11 shows the spread in the time series of annual global net atmosphere-land $CO_2$
flux and their zonally-averaged values across the eight simulations averaged over the 1997-2016
period from each simulation. The global net atmosphere-land $CO_2$ flux or net biome productivity
(NBP) is considered a critical determinant of the performance of LSMs, and is treated as such by
TRENDY, because this flux ultimately affects the changes in the atmospheric $CO_2$ burden. TRENDY
requires that LSMs simulate a terrestrial C sink for the decades of the 1990s to the present to be
considered for inclusion in the TRENDY ensemble.
Figure A11 also shows the estimates of global net atmosphere-land $CO_2$ flux from the
participating TRENDY models in grey boxes with mean and shaded ranges for the decades from
the 1960s to 2010s from the Global Carbon Project (Friedlingstein et al., 2022). Positive values in
Figure A11 indicate a C sink over land and negative values a C source to the atmosphere. In Figure
A11a, all eight simulations reported here would qualify for inclusion in the TRENDY ensemble
since they all simulate a terrestrial C sink from the 1990s to the present day. Before 1960, since
the atmospheric $CO_2$ concentration is not high enough,  the model yields both a land C sink and
source in response to interannual variability in meteorological data. In addition, the time series
of global NBP from all eight simulations lie within the uncertainty range of reported estimates
from the Global Carbon Project. Figure A11a suggests that based on global NBP, at least, it is not
possible to exclude any of the eight simulations. In Figure A11b, zonally-averaged NBP averaged
over the 1997-2016 period from each of the eight simulations mostly lie within the range of NBP
simulated by models that participated in TRENDY 2020. CLASSIC simulates a C sink at northern
high latitudes consistent with TRENDY models but it simulates a C sink on the stronger side of
TRENDY models in the southern tropics ($0°$ - $20°$S). This is likely because CLASSIC is known to
simulate low C emissions associated with LUC most of which are generated in tropical regions
(Asaadi and Arora, 2021).

Figure 9 provides additional insights into the effect of different forcings on the simulated

NBP. In Figure 9, averaged over the 1997-2016 period, an interactive N cycle leads to a somewhat
weaker C sink (panel a, 0.98 vs. 1.11 Pg C/yr), the choice of the ESA CCI based land cover leads to
a somewhat stronger C sink (panel c, 1.14 vs 0.94 Pg C/yr), and the choice of the GSWP3
meteorological data leads to a much weaker C sink (panel e, 0.74 vs 1.33 Pg C/yr) than the CRU-
JRA meteorological data. In Figure 9, panels a and b, the largest difference between the model
versions with and without the N cycle occurs in the tropics (~ 5°N - 20°S) where an interactive N
cycle leads to a weaker C sink. There are differences in zonally-averaged NBP with and without
the N cycle south of 45°S but the land area below this latitude is small so the averages are
calculated over only a few grid cells. The choice of the land cover (Figure 9, panels c and d) does
not substantially change the distribution of the zonally-averaged values of NBP although, as
noted above, the choice of ESA CCI based land cover leads to a somewhat stronger C sink. Finally,
the choice of the GSWP3 meteorological forcing leads to a weaker C sink at most latitudes (Figure
9, panels e and f).
**4.3 Automated benchmarking**
Figure 10 plots the overall score, $S_{overall}$, against benchmark scores for 16 of the 19 energy, water,
and C cycle related variables using which AMBER calculated model and benchmark scores.
AMBER does not yet evaluate N cycle related variables for which observations are more scarce
than for C cycle related variables. The range in model scores comes from the eight simulations,
and the range in benchmark scores comes from the different observation-based data sets. The
whiskers show the range in the overall score both for the benchmark and model scores. The
vertical whiskers show the range of eight model scores when a given variable from all eight model
simulations is compared to an observation-based data set. The horizontal whiskers show the
range when three or more observation-based datasets are compared to each other. When only
two observation-based data sets are compared to each other there is only one benchmark score,
and therefore there is no range. In Figure 10, three quantities are missing: soil moisture,
ecosystem respiration, and fire $CO_2$ emissions since there is only one observation-based
reference data used for these variables and therefore a benchmark score cannot be calculated.
Figure 10 shows that typically as the benchmark scores increase so do the overall model scores
or a given quantity. This indicates that uncertainty in observation-based estimates themselves
leads to a poor agreement between observations and model-simulated quantities.

For energy and water fluxes scores (panels a and b) the model overall scores lie around

the 1:1 line indicating that model scores are generally as good as the benchmark scores, except
for surface albedo (ALBS), runoff (MRRO), ground heat flux (HFG), and comparison against one
observation-based estimate of snow water equivalent which lie below the 1:1 line. For C cycle
related variables most scores lie somewhat below the 1:1 line indicating that simulated quantities
do not agree as well with observations as observations agree among themselves. The lower
benchmark score for soil C  (panel c) is because the SoilGrids250m (SG250m) data and the
Harmonized World Soil Database (HWSD) do not agree well amongst themselves because the
SG250m soil C data includes peatlands and permafrost C at high latitudes while the HWSD data
does not (see Figure 11b). Since the version of CLASSIC used here does not represent peatlands
and permafrost C it compares better with the HWSD data than with the SG250m data. In the case
of soil C, the choice of HSWD data for comparison against model values is obvious. However, for
other variables, it may not always be obvious which observation-based estimate is more
appropriate or better for comparison against model results. The uncertainty in forcing data sets
and in observation-based estimates, against which model results are evaluated, implies that even
a perfect model cannot be evaluated to its fullest extent.

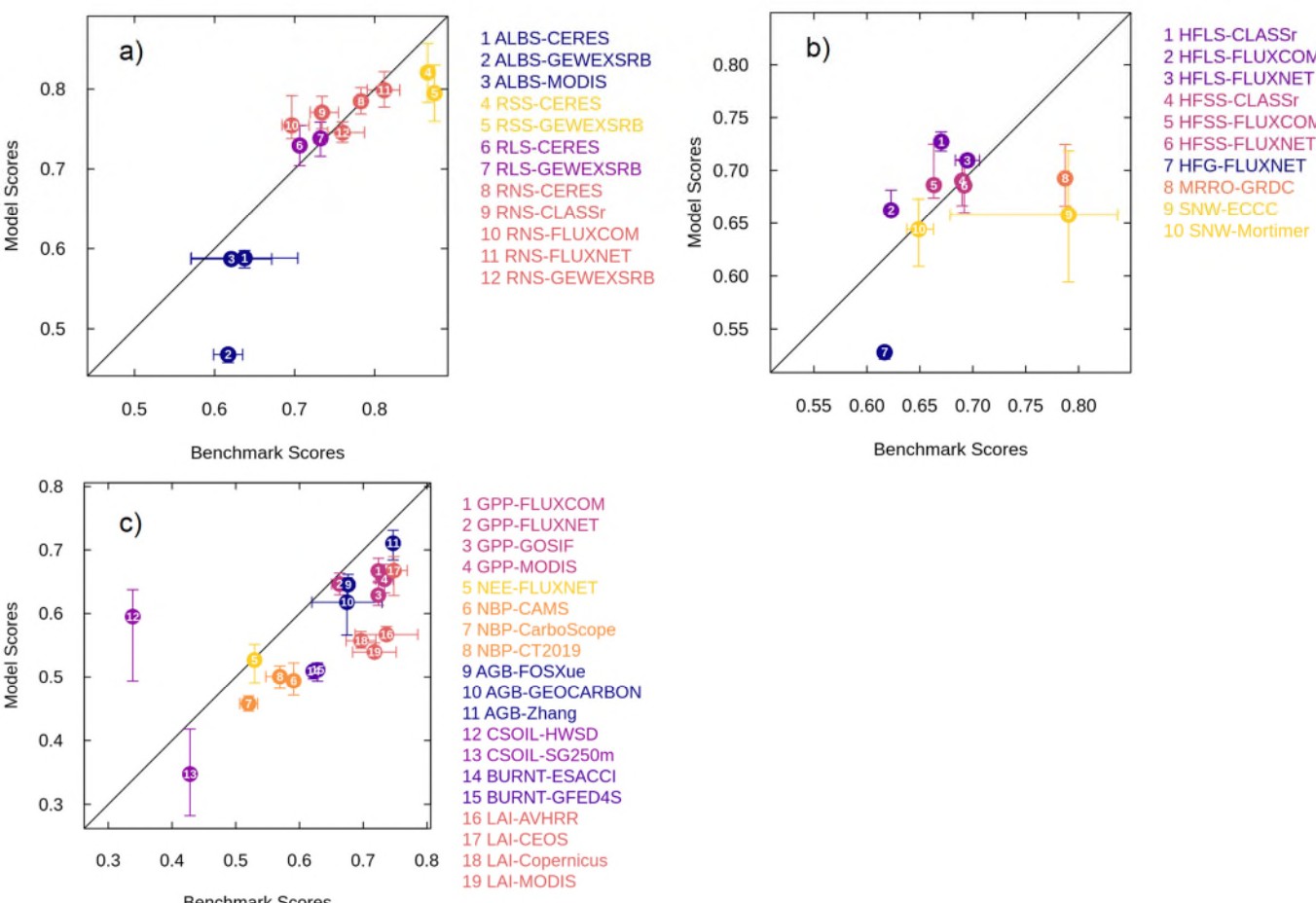

Figure 10: Comparison of benchmark scores with model overall scores for a range of energy-,
water-, and carbon-related quantities. The whiskers indicate the range for benchmark scores
across different observation-based data sets and the range across the eight model simulations
for the overall model scores. The quantities in panel (a) are ALBS (surface albedo), RSS (net
shortwave radiation), RLS (net longwave radiation), and RNS (net radiation). Quantities in panel
(b) are HFLS (latent heat flux), HFSS (sensible heat flux), HFG (ground heat flux), MRRO (runoff),
and SNW (snow water equivalent). Quantities in panel (c) are GPP (gross primary productivity),
NEE (net ecosystem exchange), NBP (net biome productivity), AGB (aboveground biomass), CSOIL
(soil carbon mass), BURNT (area burned), and LAI (leaf area index).

Figure 11 shows the zonal distribution of vegetation C mass, LAI, area burnt, GPP, and fire
$CO_2$ emissions (which constitute standard output from AMBER) and illustrates how AMBER
compares the spread across the simulations indicated by 50%, 80%, and 100% shading against
observation-based estimates. The black and shades of grey indicate the model mean and the
spread across the eight model simulations, respectively, and the thick lines in other colours show
the mean values of observation-based estimates. The time period over which observations and
model quantities are averaged is chosen to be the same. In Figure 11a, for aboveground biomass,
the GEOCARBON data set uses one product for the extratropics and another for the tropics to
create a global aboveground biomass product. The Zhang product (Zhang and Liang, 2020) is
based on the fusion of multiple gridded biomass datasets for generating a global product. Both
products are described in detail in Seiler et al. (2022). The model results generally compare better
with the Zhang product outside the 10°N to 10°S region but with the GEOCARBON product within
this region. The values to the south of 40°S are generally less reliable because of the little
vegetated land area below this latitude. In Figure 11b, the model simulated values for soil organic
C compare better with the HWSD dataset compared to the SG250m data for reasons mentioned
in the previous paragraph. Simulated leaf area index (Figure 11c) and gross primary productivity
(Figure 11e) generally compare well their observation-based estimates. The simulated area
burned (Figure 11d) and fire emissions (Figure 11f) also compare well with observation-based
estimates except that the model is not able to capture the small area burned and emissions at
northern high latitudes between around 50°N to 70°N. Figures A12 and A13 compare zonally
averaged values of other simulated quantities with observation-based estimates used in the
AMBER framework. Together Figures 11, A12, and A13  illustrate that the model is overall able
to capture the latitudinal distribution of most land surface quantities.

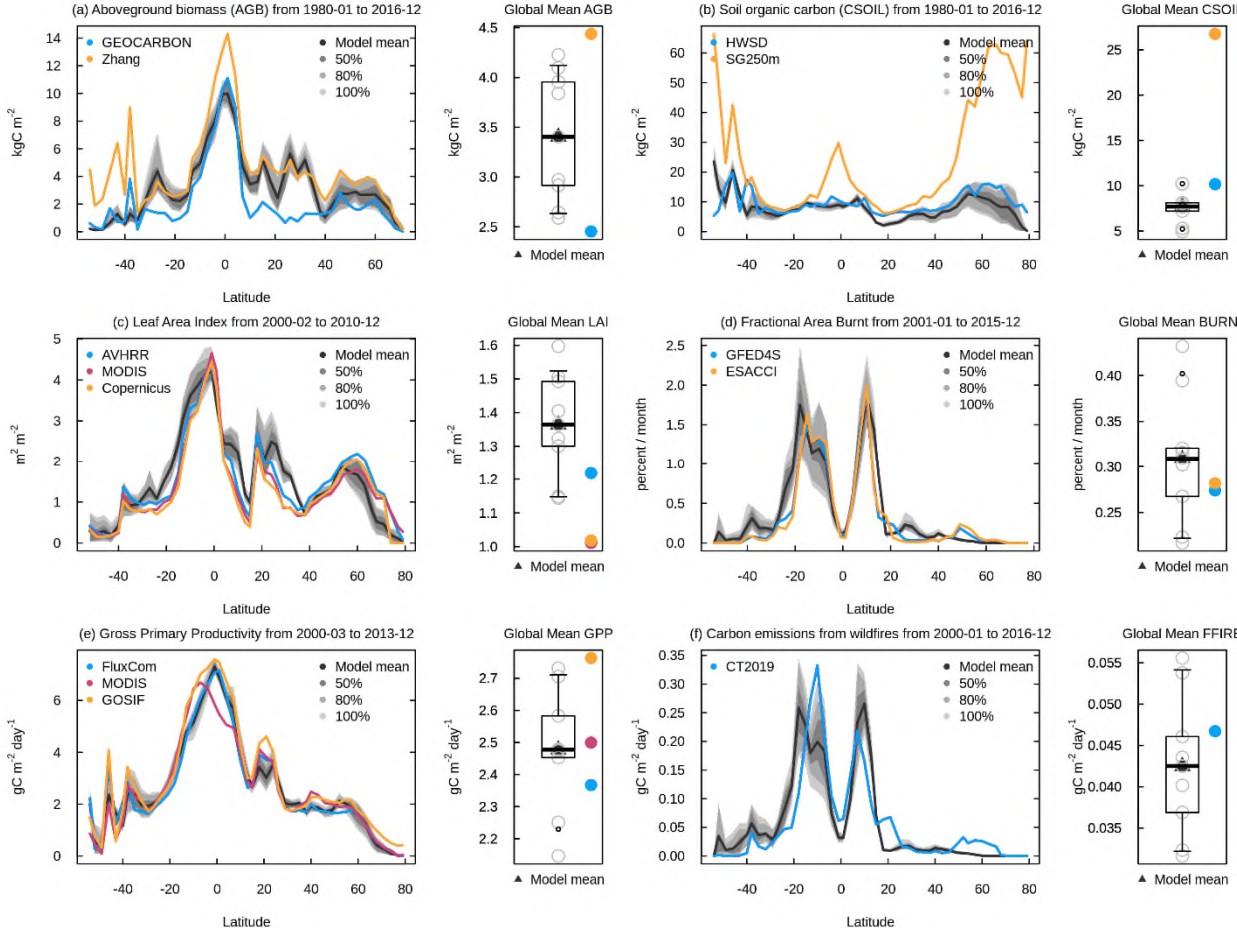


Figure 11: Zonally-averaged values of aboveground biomass (a), soil carbon mass (b), leaf area
index (c), fractional area burnt (d), gross primary productivity (e), and fire $CO_2$ emissions (f) from
the eight simulations summarized in Table 1. The model results are shown as their mean (black)
and the spread across the eight simulations indicated by 50%, 80%, and 100% ranges in different
shades of grey. The observation-based estimates used in AMBER to calculate scores are shown
in coloured lines.

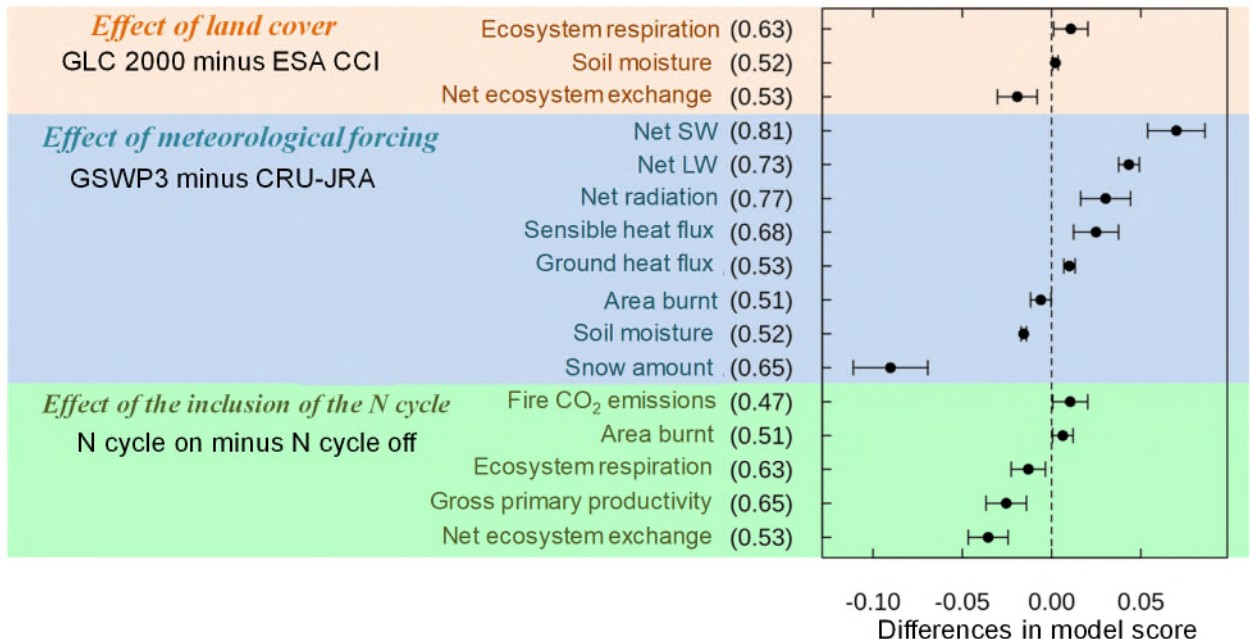

Figure 12: Summary of difference in overall scores for model simulated quantities and
combinations for which the differences are statistically significant. The scores in parentheses for
each quantity are the average scores across the eight simulations and provide context. The error
bars denote the 95% confidence interval as explained in the text.

Since overall scores are available for all eight simulations for model quantities that are
compared to observations it is possible to evaluate how an interactive N cycle, and the choice of
meteorological data and land cover data affect model performance. Figure 12 summarizes the
difference in overall scores for model quantities and combinations for which the differences are
statistically significant at the 5% level based on Tukey's test (Tukey, 1977). The score indicated in
parentheses for each quantity is the average score across the eight simulations and provides
context. For example, when evaluating the effect of change in land cover for NEE the use of the
GLC 2000 based land cover, compared to the use of the ESA CCI based land cover, degrades the
average score for net ecosystem exchange by about 0.02 given that the average score for net
ecosystem exchange in 0.53. The error bars on the value 0.02 denote the 95% confidence interval
and in this case are calculated by differencing four simulations that use the GLC 2000 based land
cover versus four simulations that use the ESA CCI based land cover. The use of the GLC 2000
based land cover on the other hand slightly improves scores for ecosystem respiration and liquid
soil moisture. The use of GSWP3 data improves model scores for net shortwave, longwave, and
total radiation, for sensible and ground heat flux but degrades the overall score for area burned,
soil moisture, and more so for snow water equivalent. Finally, an interactive N cycle slightly
improves model performance for area burned and fire $CO_2$ emissions (due to improved
aboveground biomass in the tropics) but degrades it for ecosystem respiration, GPP, and net
ecosystem exchange. The inclusion of an interactive N cycle changes $V_{c,max}$ to a prognostic
variable for each PFT as opposed to being specified based on observations. This is analogous to
running an atmospheric model with a fully dynamic 3-dimensional ocean as opposed to using
specified sea surface temperatures (SST) and sea ice concentrations (SIC). Using a dynamic ocean
allows future projections (since future SSTs and SICs are not known) but invariably degrades a
model's performance for the present day since simulated SSTs and SICs will have their biases.
Similarly, using an interactive N cycle allows to project future changes in $V_{c,max}$ (based on changes
in N availability) but also degrades CLASSIC's performance for the present day since simulated
$V_{c,max}$ has its own biases. Overall, the model performance is most affected by the choice of the
driving meteorological data for water and energy fluxes, and by the inclusion or absence of an N
cycle and by the choice of land cover for carbon-cycle related state variables and fluxes.


## 5. Conclusions


The response of the terrestrial biosphere over the historical period is driven primarily by
four global change drivers – increasing atmospheric $CO_2$, changing climate, LUC, and N deposition
and fertilizer application. Our framework allows us to evaluate how a land surface model
responds to increasing atmospheric $CO_2$, changing climate, and anthropogenic N additions to the
coupled soil-vegetation system and how this response is dependent on two driving
meteorological data sets, two land cover representations, and the two model variations (with
and without an interactive N cycle). However, the framework used here does not quantify the
uncertainty associated with LUC over the historical period since we use only one reconstruction
of increasing crop area over the historical period. These results help draw three primary
conclusions. First, even if the observations and models were perfect (including their structure
and their parameterizations) the uncertainty associated with driving meteorological data and
geophysical fields makes it difficult to evaluate LSMs. The uncertainty in global scale driving data
implies that a model can never be truly evaluated to its fullest extent. Model results can only be
as good as the data that are used to force them and therefore even a perfect model cannot yield
perfect results.
Second, model tuning when driving the model with a single set of forcings and evaluating
it against a single set of observations is likely not a fruitful exercise. Models should not be tuned
to a single set of driving data and observation-based evaluation data. Rather their performance
must be evaluated against a range of available observations in light of the uncertainty associated
with driving data and the uncertainty associated with observations. A model's ability to
reproduce a given single set of observations when driven with a single set of driving data is not a
true measure of its success. Here again, a perfect model driven by perfect forcing data cannot be
truly evaluated to its fullest extent since observations themselves have uncertainties.

Third, with the caveat that our framework uses only one reconstruction of increase in

crop area over the historical period, the response of a model expressed in terms of net
atmosphere-land $CO_2$ flux to perturbation in meteorological, $CO_2$, and LUC forcing over the
historical period appears to be largely independent of its pre-industrial state as simulated here.
The pre-industrial soil and vegetation C mass for the eight simulations considered here vary
between 1035 ± 195 Pg C and 405 ± 58 Pg C (mean ± standard deviation), respectively. Both pre-
industrial and present-day vegetation and soil C pools explain only about 2% to 7% of the
variability in simulated net atmosphere-land $CO_2$ flux (Figure A11) over the 1997-2016 period of
each of the eight simulations. The net atmosphere-$CO_2$ flux from all eight simulations for the
period the 1960s to 2000s is found to lie within the uncertainty range provided by the GCP
(Friedlingstein et al., 2022). Given the current uncertainty in net atmosphere-land $CO_2$ flux, it is
therefore not possible to exclude any of the eight simulations at least on this basis. The finding
that a transient response of a model is independent of its pre-industrial state is also consistent
with land components of CMIP6 models. Arora et al. (2020) analyzed results from CMIP6
simulations in which atmospheric $CO_2$ increases at a rate of 1% per year from the year 1850 until
$CO_2$ quadruples from ~285 to ~1140 ppm. They found that the C-concentration and C-climate
feedback parameters for the land component of CMIP6 models do not depend on the absolute
values of their vegetation and soil C pools but rather how a given model responds to changes in
atmospheric $CO_2$ and the associated change in temperature. This conclusion is perhaps
somewhat comforting in that while pre-industrial states of LSMs may be different from their true
observed states they still have the ability to reproduce net atmosphere-land $CO_2$ flux over the
historical period that is consistent with current observation-based estimates. Clearly, this
reasoning does not apply if pre-industrial vegetation or soil C mass are zero. One reason why
present day net atmosphere-land $CO_2$ flux is independent of a LSM's pre-industrial state is
because the model is first spun up to equilibrium conditions and then forced with time-variant
forcings. However, successful reproduction of atmosphere-land $CO_2$ fluxes over the historical
period is no guarantee that future projections from LSMs are reliable.

The ensemble-based approach used here also allows for the evaluation of the effect of a

given meteorological forcing and land cover, and the effect of an interactive N cycle on model
simulated quantities in a robust manner. Ensemble averages of simulations that use the CRU-JRA
and GSWP3 meteorological forcing show that the use of the GSWP3 meteorological forcing yields
lower evapotranspiration (latent heat flux), higher runoff, higher sensible heat flux, a higher
burned area, and a weaker land C sink for the present day compared to when the CRU-JRA
meteorological forcing is used. Possible reasons that explain these differences when using the
GSWP3 meteorological data are the higher frequency of high precipitation events (greater than
~5-10 mm/day) (Figure A2) and 0.93 °C higher temperature in the northern tropical region (Figure
A1h) in the GSWP3 compared to the CRU-JRA meteorological data. High precipitation intensity in
regions of high annual precipitation (e.g. the tropical regions) would lead to more surface runoff
since less precipitation infiltrates the top soil layer, further leading to less soil moisture, less
evapotranspiration, higher sensible heat flux, and more area burned. Higher temperatures in the
northern tropical region in the GSWP3 meteorological data certainly contribute to all these
differences (except higher runoff). While, annual globally-averaged soil moisture is about 4%
higher in the simulations driven with the GSWP meteorological data (Figure 2c), in several parts
of the tropical regions annual simulated soil moisture is lower for GSWP3 simulations (not
shown). The use of the ESA CCI land cover leads to higher soil C, higher GPP, and higher area
burned primarily because of the larger grass area when land cover is based on the ESA CCI
product compared to the GLC 2000 product. The use of the ESA CCI based land cover also leads
to a slightly weaker land C sink for the present day. Finally, the comparison of simulations with
and without the N cycle averaged over all meteorological data and land cover combinations
allows us to identify the effect of the N cycle. Simulated vegetation C mass and GPP are lower in
the model version with the interactive N cycle. In particular, we found that the somewhat low
productivity at high latitudes, when the N cycle is turned on, leads to relatively large differences
in soil C at high latitudes regardless of the meteorological data or land cover being used to drive
the model. Although, this is not the reason for differences in net atmosphere-land $CO_2$ flux
between models with and without N cycling: as mentioned above present-day net atmosphere-
land $CO_2$ flux is independent of both the pre-industrial and present-day vegetation and soil C
pools. Given the knowledge about the effect of N cycling on model behaviour, the reasons can
now be investigated to further improve the N cycle component of CLASSIC.

It is logical to assume that the results presented here are sensitive to the horizontal

resolution of the model. Both forcing data that are used to drive the model, and observations
against which model results are compared, are regridded to be consistent with the model's
spatial resolution. For example, at the scale of a few meters, meteorological variables measured
at a given site will indeed be less uncertain than their spatially-averaged values say for a 2.81°
grid cell. Similarly, observations at a scale of a few meters for soil C and/or vegetation C mass will
also likely be more certain than their values at large spatial scales. This is one reason why AMBER
uses both gridded and in-situ observation-based estimates to calculate its scores. Fluxes of latent
and sensible heat, on the other hand, may not be any more certain at a given site than over large
spatial scales. This is because of the problems associated with energy budget closure (Mauder et
al., 2020) which, at the point scale, prevent the sum of annual latent and sensible heat flux to be
equal to net radiation (average of ground heat fluxes is close to zero at an annual time scale).

LSMs have become increasingly complex over the years and so has the requirement for

forcing data to drive these models. The evaluation of LSMs has also become complex as the
models now generate a multitude of variables that must be evaluated against their observation-
based estimates. Estimates of observation-based data to evaluate models, and the availability of
forcing data, have also increased. Given the uncertainties associated with model inputs, model
structure, and observation-based data, it is unrealistic to expect LSMs to perfectly reproduce
observations for large-scale global simulations. It is not known *a priori* which model structure,
forcing data sets, and observation data sets are better. Driving data including meteorological data
sets and land cover representations may be more realistic in some parts of world and less in
others. Observation-based data sets also have their limitations and attributes which may make
them better or ill-suited for comparison with a given model.  A more robust model evaluation
must therefore take into account the uncertainties both in the forcing and observation-based
data. A comprehensive and robust model evaluation can be performed by comparing multiple
model realizations against multiple observation-based data sets.

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

**Author contribution**
VA and SKG performed the simulations, and VA wrote the majority of the manuscript. CS performed the
AMBER related analysis. LW put together the ESA CCI land cover. CS, LW, and SKG provided comments
on the entire manuscript and also wrote their respective sections.

**Competing interests**
There are no competing interests.

**Acknowledgment**

We thank Joe Melton for providing comments on an earlier version of this paper. We also thank
Benjamin Bond-Lamberty for taking this paper on as an Associate Editor, and the two anonymous
reviewers for providing helpful comments which greatly improved this paper.

# Appendix

**A1: Automated Model Benchmarking R Package (AMBER)**

The Automated Model Benchmarking R package quantifies model performance using five scores
that assess a model's bias ($S_{bias}$), root-mean-square-error ($S_{rmse}$), seasonality ($S_{phase}$), inter-annual
variability ($S_{iav}$), and spatial distribution ($S_{dist}$). All scores are dimensionless and range from zero
to one, where increasing values imply better performance. The exact definition of each skill score
is provided below.
**A1.1 Bias Score ($S_{bias}$)**
The bias is defined as the difference between the time-mean values of model and reference data:
$$bias(\lambda, \phi) = \overline{v_{mod}}(\lambda, \phi) - \overline{v_{ref}}(\lambda, \phi), \quad\quad\quad\quad\text{(A1)}$$
where $\overline{v_{mod}}(\lambda, \phi)$ and $\overline{v_{ref}}(\lambda, \phi)$ are the mean values in time ($t$) of a variable $v$ as a function of
longitude $\lambda$ and latitude $\phi$ for model and reference data, respectively. Nondimensionalization is
achieved by dividing the bias by the standard deviation of the reference data ($\sigma_{ref}$):
$$\varepsilon_{bias}(\lambda, \phi) = \frac{|bias(\lambda,\phi)|}{\sigma_{ref}(\lambda,\phi)} \quad\quad\quad\quad\text{(A2)}$$
Note that $\varepsilon_{bias}$ is always positive, as it uses the absolute value of the bias. For evaluations against
stream flow measurements, the bias is divided by the annual mean rather than the standard
deviation of the reference data. This is because we assess streamflow on an annual rather than
monthly basis, implying that the corresponding standard deviation is small. The same approach
is applied to soil C and vegetation C mass, whose reference data provide a static snapshot in time.
For both of these cases, $\varepsilon_{bias}(\lambda, \phi)$ becomes:
$$\varepsilon_{bias}(\lambda, \phi) = \frac{|bias(\lambda, \phi)|}{\overline{v_{ref}(\lambda, \phi)}}$$
(A3)


A bias score that ranges from zero to one is calculated next:
$$s_{bias}(\lambda, \phi) = e^{-\varepsilon_{bias}(\lambda, \phi)}$$
(A4)

While small relative errors yield score values close to one, large relative errors cause score values
to approach zero. Taking the mean of $s_{bias}$ across all latitudes and longitudes, denoted by a double
bar over a variable, leads to the scalar score:
$$S_{bias} = \overline{\overline{s_{bias}(\lambda, \phi)}}$$
(A5)


**A1.2 Root-Mean-Square-Error Score ($S_{rmse}$)**
While the bias assesses the difference between time-mean values, the root-mean-square-error
(*rmse*) is concerned with the residuals of the modeled and observed time series:
$$rmse(\lambda, \phi) = \sqrt{\frac{1}{t_f - t_0} \int_{t_0}^{t_f} \left( v_{mod}(t, \lambda, \phi) - v_{ref}(t, \lambda, \phi) \right)^2 dt}$$
(A6)


where $t_0$ and $t_f$ are the initial and final time steps, respectively. A similar metric is the centralized
*rmse* (*crmse*), which is based on the residuals of the anomalies:

$crmse(\lambda, \phi) = \sqrt{\frac{1}{t_f - t_0} \int_{t_0}^{t_f} \left[ \left( v_{mod}(t, \lambda, \phi) - \overline{v_{mod}}(\lambda, \phi) \right) - \left( v_{ref}(t, \lambda, \phi) - \overline{v_{ref}}(\lambda, \phi) \right) \right]^2 dt}$ (A7)

The *crmse*, therefore, assesses residuals that have been bias-corrected. Since we already
assessed the model's bias through $S_{bias}$, it is convenient to assess the residuals using *crmse* rather
than *rmse*. In a similar fashion to the bias, we then compute a relative error:
$$\varepsilon_{rmse}(\lambda, \phi) = \frac{crmse(\lambda,\phi)}{\sigma_{ref}(\lambda,\phi)} \tag{A8}$$

scale this error onto a unit interval:
$$s_{rmse}(\lambda, \phi) = e^{-\varepsilon_{rmse}(\lambda,\phi)} \tag{A9}$$

and compute the spatial mean:
$$S_{rmse} = \overline{\overline{s_{rmse}(\lambda, \phi)}} \tag{A10}$$

**A3 Phase Score ($S_{phase}$)**
The skill score $S_{phase}$ assesses how well the model reproduces the seasonality of a variable by
computing the time difference $\theta(\lambda, \phi)$ between modeled and observed month of maxima of the
climatological mean cycle:
$$\theta(\lambda, \phi) = \text{maxima}\big(c_{mod}(t, \lambda, \phi)\big) - \text{maxima}\Big(c_{ref}(t, \lambda, \phi)\Big) \tag{A11}$$

where $c_{mod}$ and $c_{ref}$ are the climatological mean cycle of the model and reference data,
respectively. The operator *maxima* in equation A11 calculates the month in which the maximum
of a given quantity occurs. The time difference $\theta(\lambda, \phi)$ in months is then scaled from zero to one
based on the consideration that the maximum possible time difference is 6 months:
$$s_{phase}(\lambda, \phi) = \frac{1}{2}\left[1 + \cos\left(\frac{2\pi \, \theta(\lambda,\phi)}{12}\right)\right] \tag{A12}$$

The spatial mean of $s_{phase}$ then leads to the scalar score:
$$S_{phase} = \overline{\overline{s_{phase}(\lambda, \phi)}}$$
(A13)


**A4 Inter-Annual Variability Score ($S_{iav}$)**
The skill score $S_{iav}$ quantifies how well the model reproduces patterns of inter-annual variability.
This score is based on data where the seasonal cycle ($c_{mod}$ and $c_{ref}$) has been removed:
$$iav_{mod}(\lambda, \phi) = \sqrt{\frac{1}{t_f - t_0} \int_{t_0}^{t_f} \left(v_{mod}(t, \lambda, \phi) - c_{mod}(t, \lambda, \phi)\right)^2 dt}$$
(A14)

$$iav_{ref}(\lambda, \phi) = \sqrt{\frac{1}{t_f - t_0} \int_{t_0}^{t_f} \left(v_{ref}(t, \lambda, \phi) - c_{ref}(t, \lambda, \phi)\right)^2 dt} \; .$$
(A15)


The relative error, nondimensionalization, and spatial mean are computed next:
$$\varepsilon_{iav}(\lambda, \phi) = \left|iav_{mod}(\lambda, \phi) - iav_{ref}(\lambda, \phi)\right| / iav_{ref}(\lambda, \phi)$$
(A16)

$$s_{iav}(\lambda, \phi) = e^{-\varepsilon_{iav}(\lambda, \phi)}$$
(A17)

$$S_{iav} = \overline{\overline{s_{iav}(\lambda, \phi)}}$$
(A13)

**A5 Spatial Distribution Score ($S_{dist}$)**
The spatial distribution score $S_{dist}$ assesses how well the model reproduces the spatial pattern of
a variable. The score considers the correlation coefficient $R$ and the relative standard deviation $\sigma$
between $\overline{v_{mod}}(\lambda, \phi)$ and $\overline{v_{ref}}(\lambda, \phi)$. The score $S_{dist}$ increases from zero to one, the closer $R$ and
$\sigma$ approach a value of one. No spatial integration is required as this calculation yields a single
value:
$$S_{dist} = 2(1 + R)\left(\sigma + \frac{1}{\sigma}\right)^{-2}$$
(A19)

where $\sigma$ is the ratio between the standard deviation of the model and reference data:
$$\sigma = \sigma_{\overline{v_{mod}}} / \sigma_{\overline{v_{ref}}}$$
(A20)

and $\sigma_{\overline{v_{mod}}}$ and $\sigma_{\overline{v_{ref}}}$ are the standard deviations of the annual mean values from the model and
reference/observation-based data, respectively, and therefore are scalars.
**A6 Overall Score ($S_{overall}$)**
As a final step, scores are averaged to obtain an overall score:
$$S_{overall} = \frac{S_{bias} + 2\,S_{rmse} + S_{phase} + S_{iav} + S_{dist}}{1+2+1+1+1}$$
(A21)

Note that $S_{rmse}$ is weighted by a factor of two and is an entirely subjective decision but follows
Collier et al. (2018).

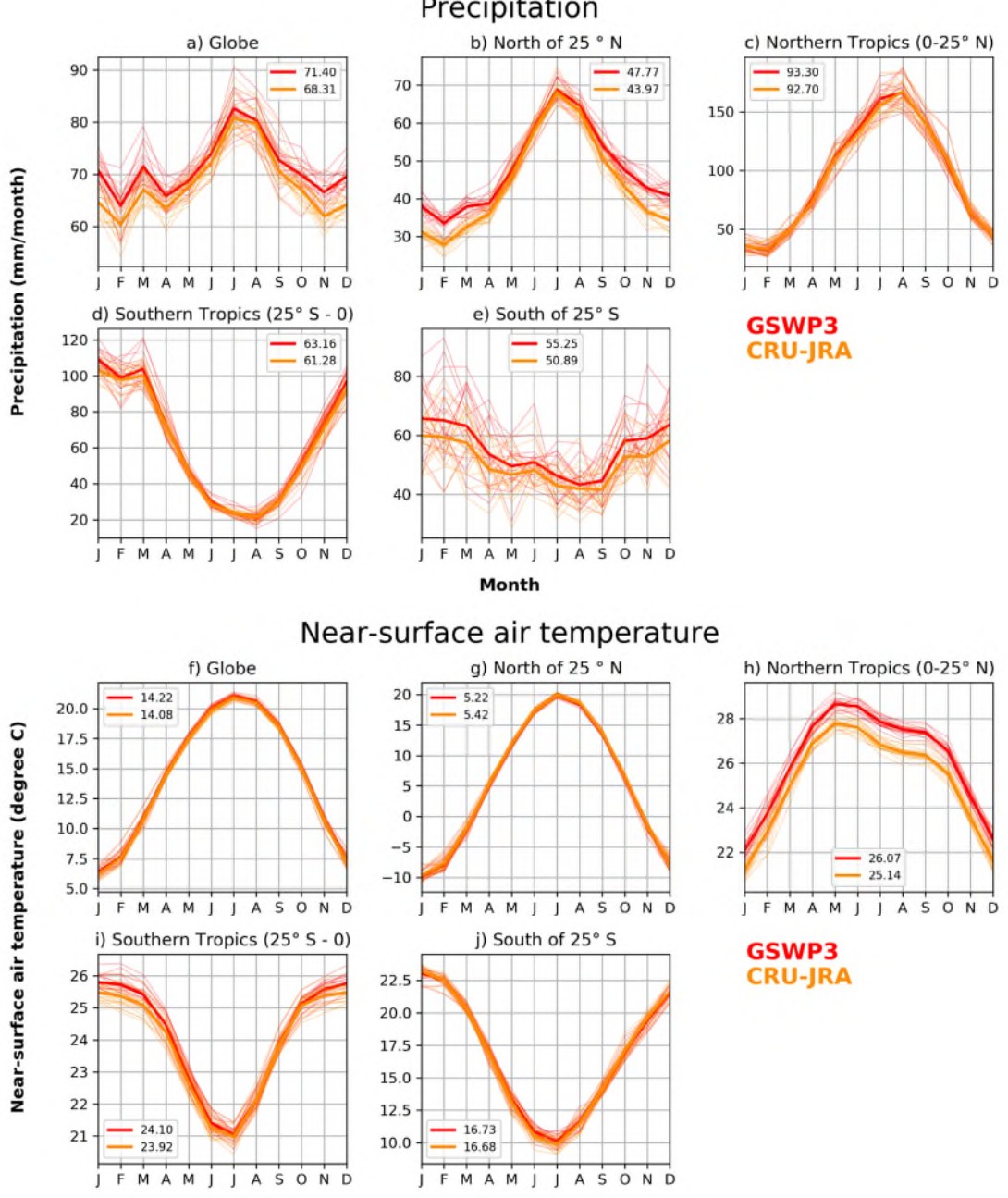

Figure A1: Comparison of monthly precipitation (upper panel) and temperature (lower panel) for five global regions (global, north of 25 °N, northern and southern tropics, and south of 25 °S) from the CRU-JRA and GSWP3 meteorological forcing data sets that are used to drive the CLASSIC model. The global and regional averages exclude Greenland and Antarctica. The legend entries show the annual mean values averaged over the 1997-2016 period. The thin lines show individual years and the thick line is their average.

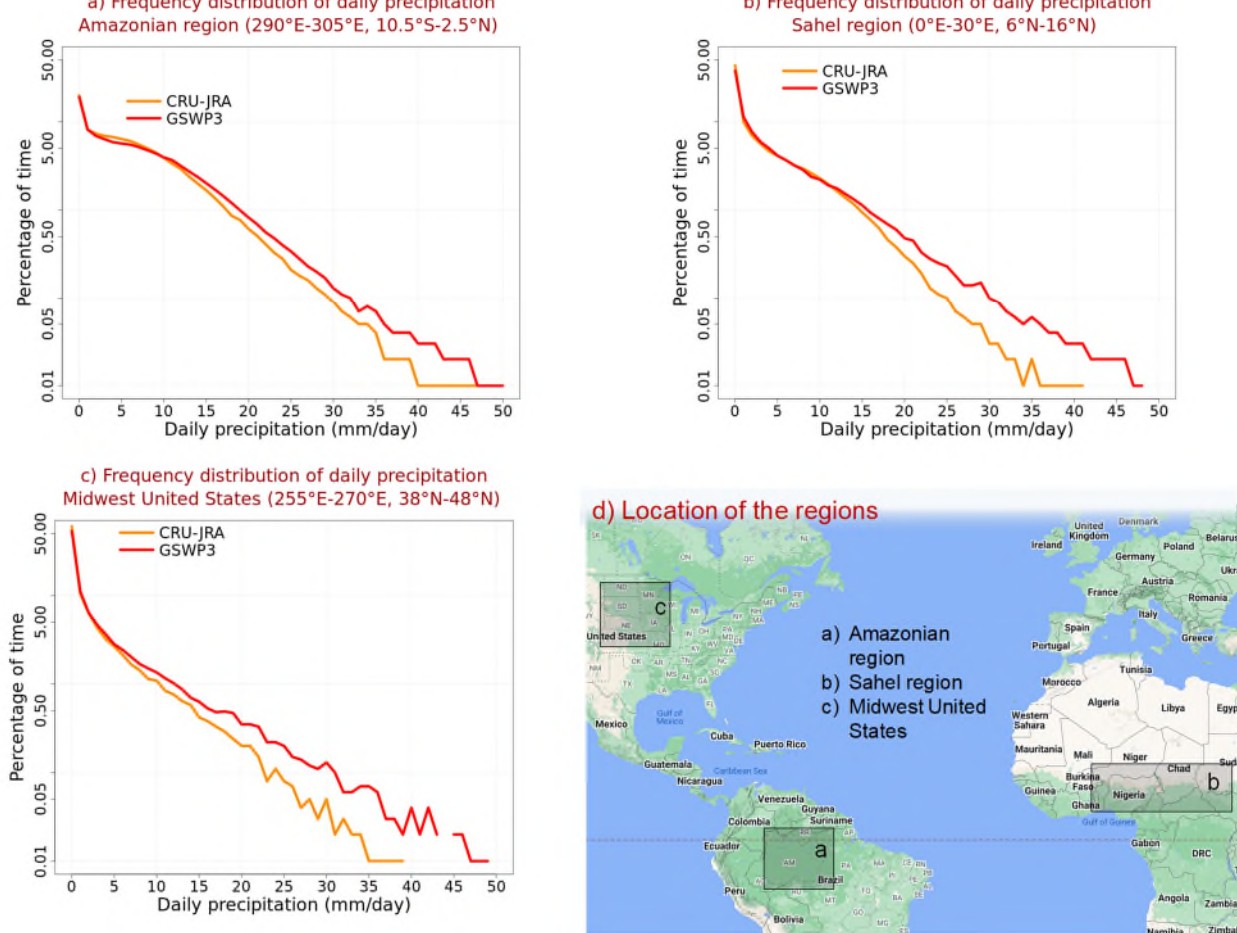

Figure A2: Comparison of the frequency distribution of daily precipitation between the CRU-JRA
and GSWP3 meteorological data sets for three broad regions and the period 1997-2016: a) the
Amazonian region, b) the Sahel region, and c) the Midwest United States. The frequency is
represented as a percentage of time daily precipitation is between *x* and *x*+1 mm/day, where x
is the value on the x-axis. Panel (d) shows the location of these broad regions. The underlying
map in panel (d) is from Google Maps.



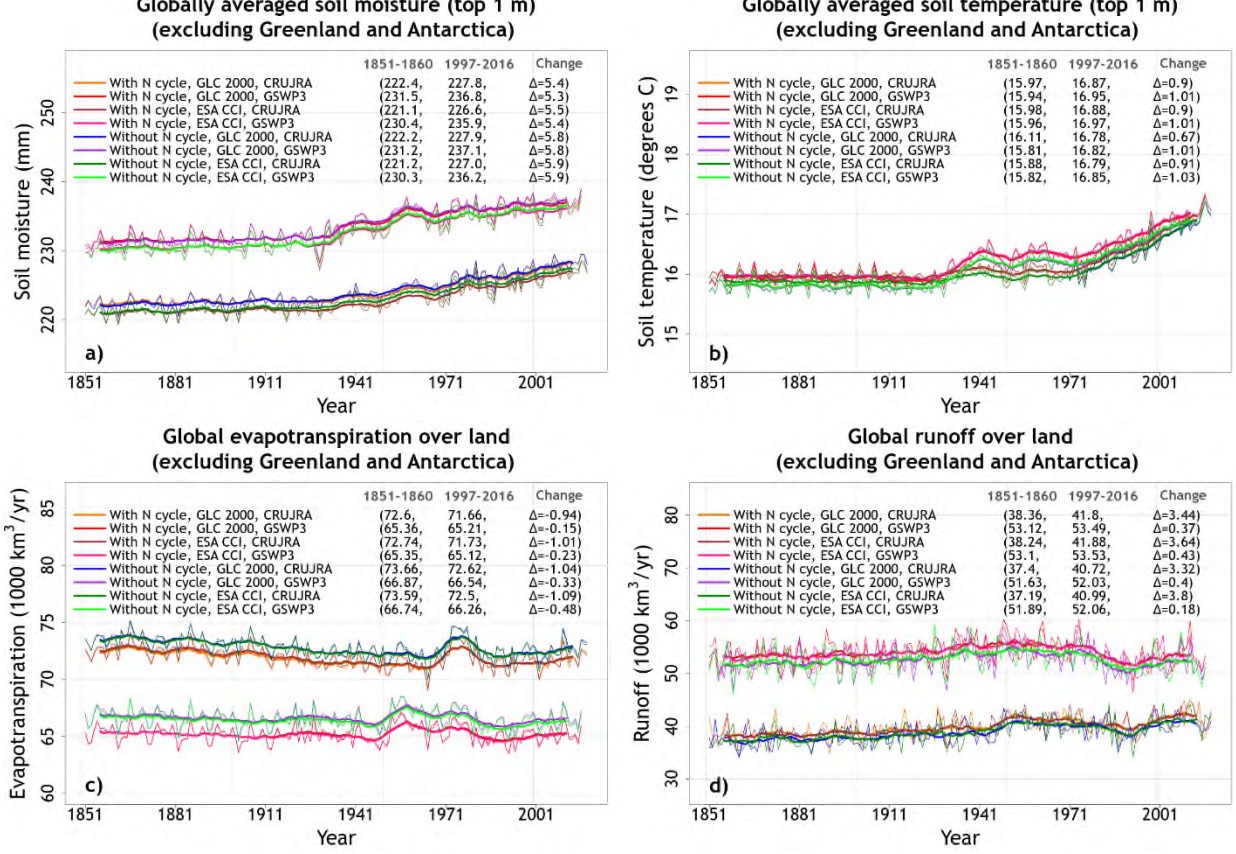


Figure A3: Time series of simulated globally-averaged annual soil moisture (a) and soil
temperature (b) in the top 1m, global annual evapotranspiration (c), and runoff (d) from the
eight simulations summarized in Table 1. The thin lines show the individual years and the thick
lines show their 11-year moving average. Model values averaged over the pre-industrial (1851-
1860) and present-day (1997-2016) time periods, and their difference, are also shown.



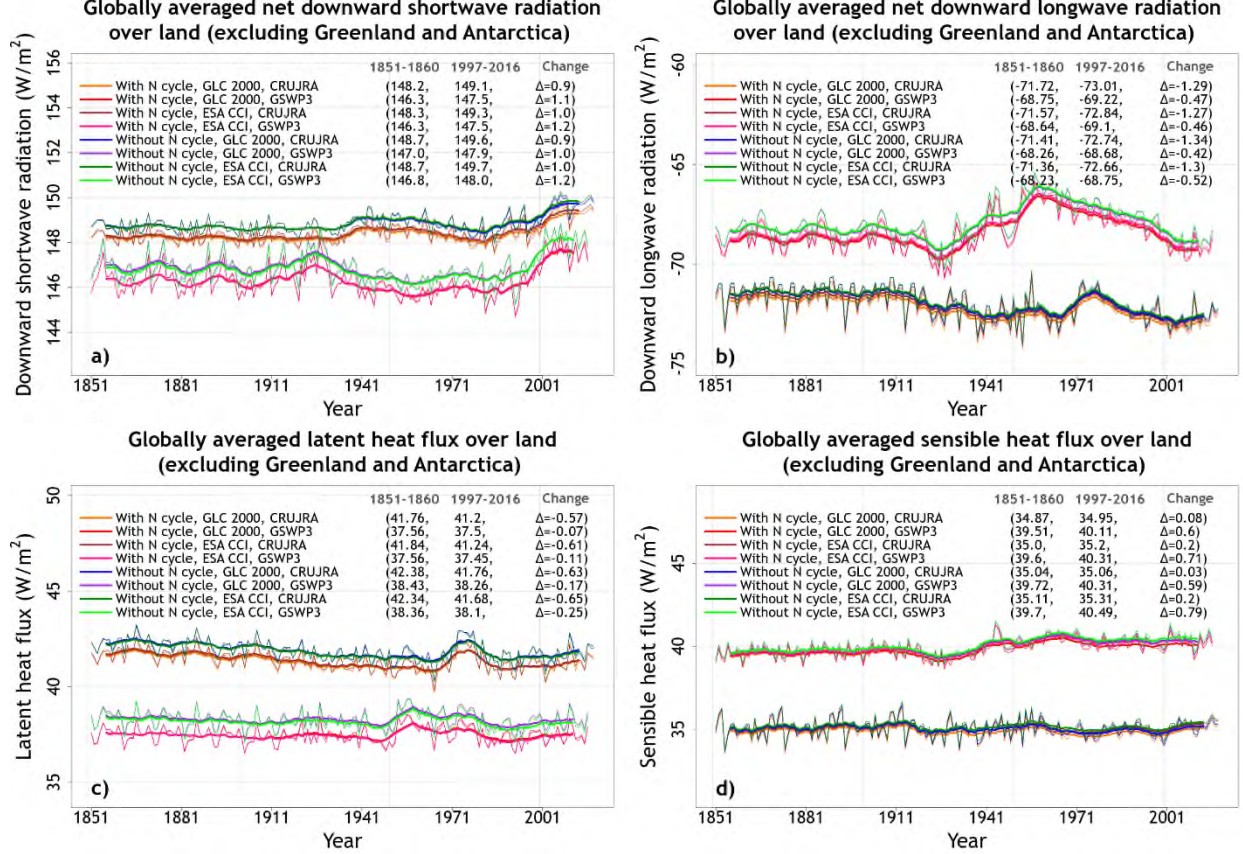


Figure A4: Time series of simulated globally-averaged annual energy fluxes from the eight
simulations summarized in Table 1. Panel (a) shows net downward shortwave radiation, panel
(b) shows net downward longwave radiation, panel (c) shows latent heat flux, and panel (d)
shows sensible heat flux. The thin lines show the individual years and the thick lines show their
11-year moving average. Model values averaged over the pre-industrial (1851-1860) and
present-day (1997-2016) time periods, and their difference, are also shown for individual
simulations.




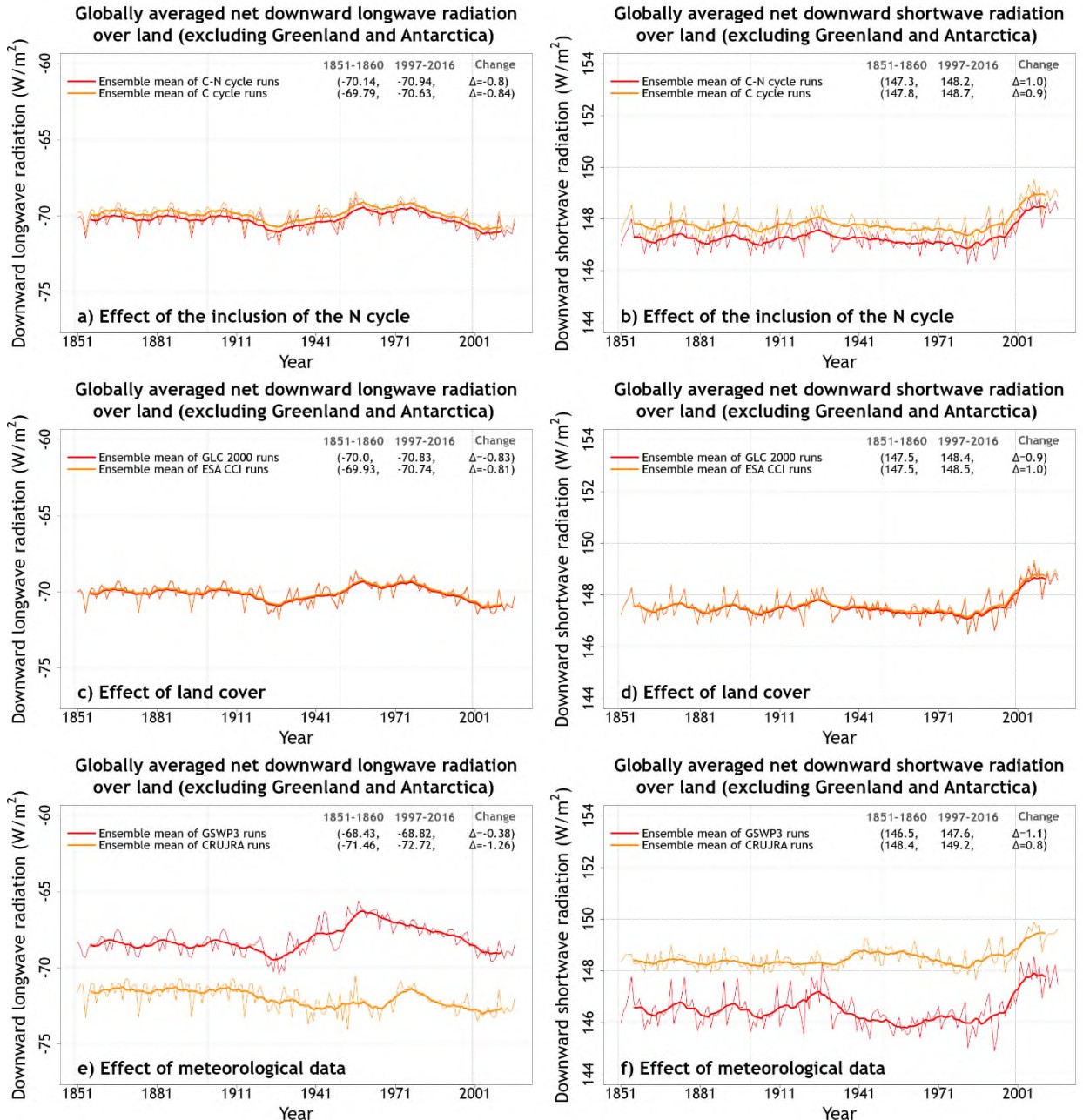


Figure A5: Time series of globally-averaged annual net downward longwave and shortwave
radiation (over all land area excluding Greenland and Antarctica) averaged over the four
ensemble members each that are driven with and without N cycle (panels a, b), driven with GLC
2000 and ESA CCI based land cover (panels c, d), and driven with GSWP3 and CRU-JRA
meteorological data (panels e, f). The thin lines show the individual years and the thick lines show
their 11-year moving average. Model values averaged over the pre-industrial (1851-1860) and
present-day (1997-2016) time periods, and their difference, are also shown.

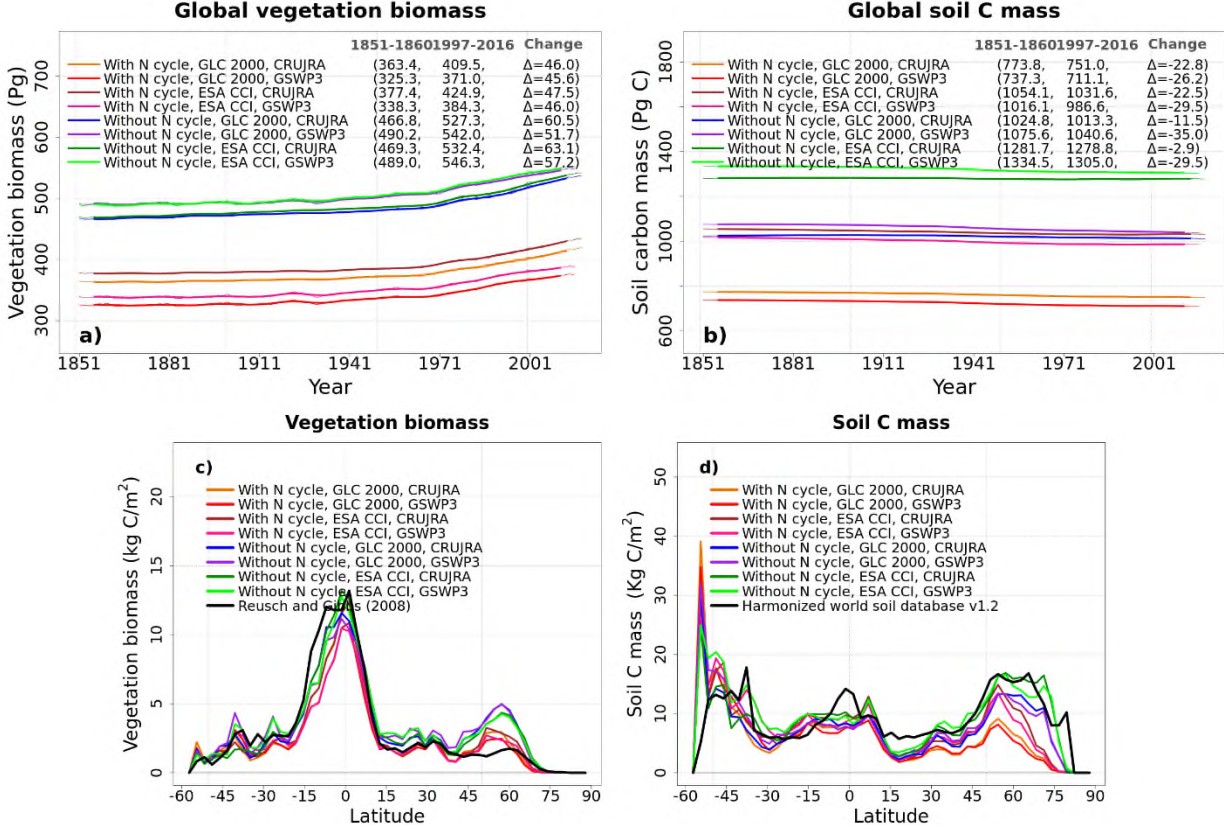

Figure A6: Time series of simulated global annual vegetation carbon mass (a) and soil carbon (b) from the eight simulations summarized in Table 1. The global totals exclude Greenland and Antarctica. Panels (c) and (d) show the zonally-averaged values of vegetation carbon mass and soil carbon mass over land from the eight simulations averaged over the 1997-2016 period. The thin lines show the individual years and the thick lines show their 11-year moving average in panels (a) and (b). Model values averaged over the pre-industrial (1851-1860) and present-day (1997-2016) time periods, and their difference, are also shown in panels (a) and (b).



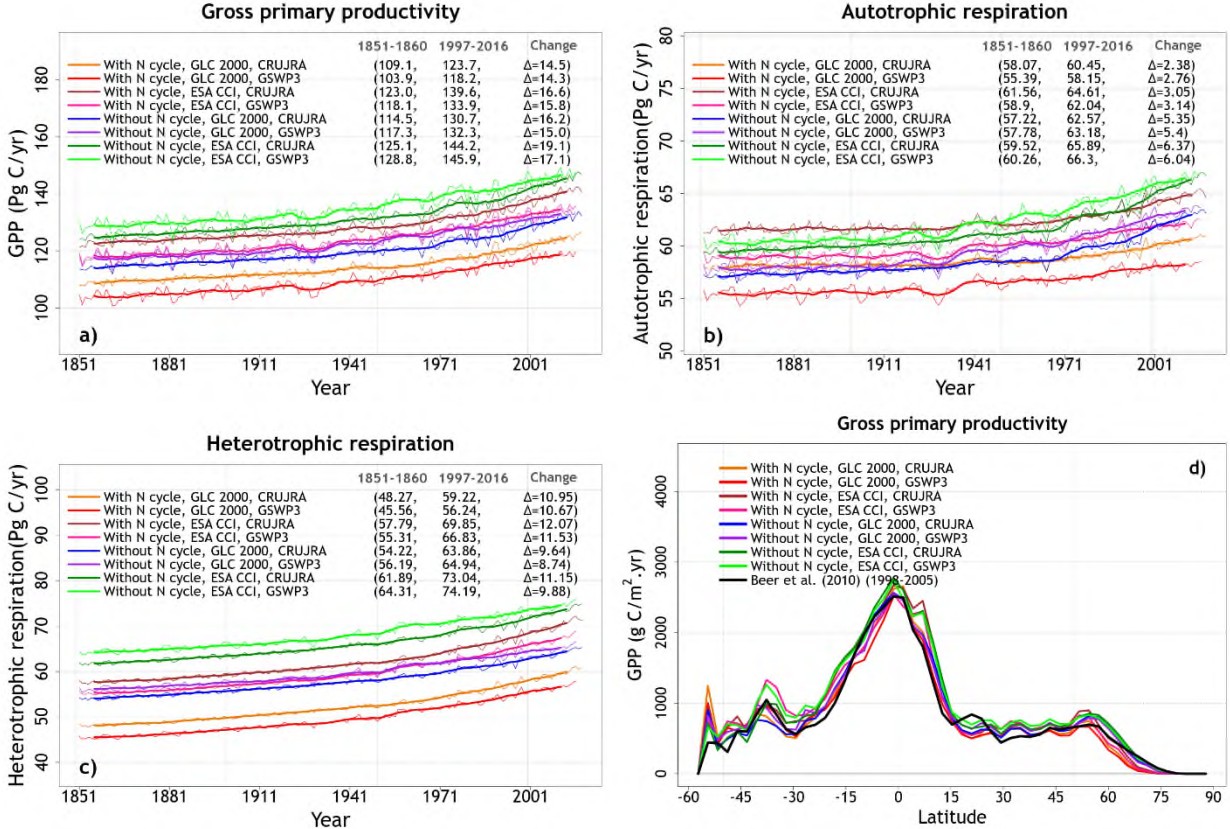


Figure A7: Time series of simulated global annual gross primary productivity (GPP) (a),
autotrophic respiration (b), and heterotrophic respiration (c) from the eight simulations
summarized in Table 1. Panel (d) shows the zonally-averaged values of GPP from the eight
simulations averaged over the 1997-2016 period for each simulation. The thin lines show the
individual years and the thick lines show their 11-year moving average in panels (a) to (c). Model
values averaged over the pre-industrial (1851-1860) and present-day (1997-2016) time periods,
and their difference, are also shown in panels (a) to (c).



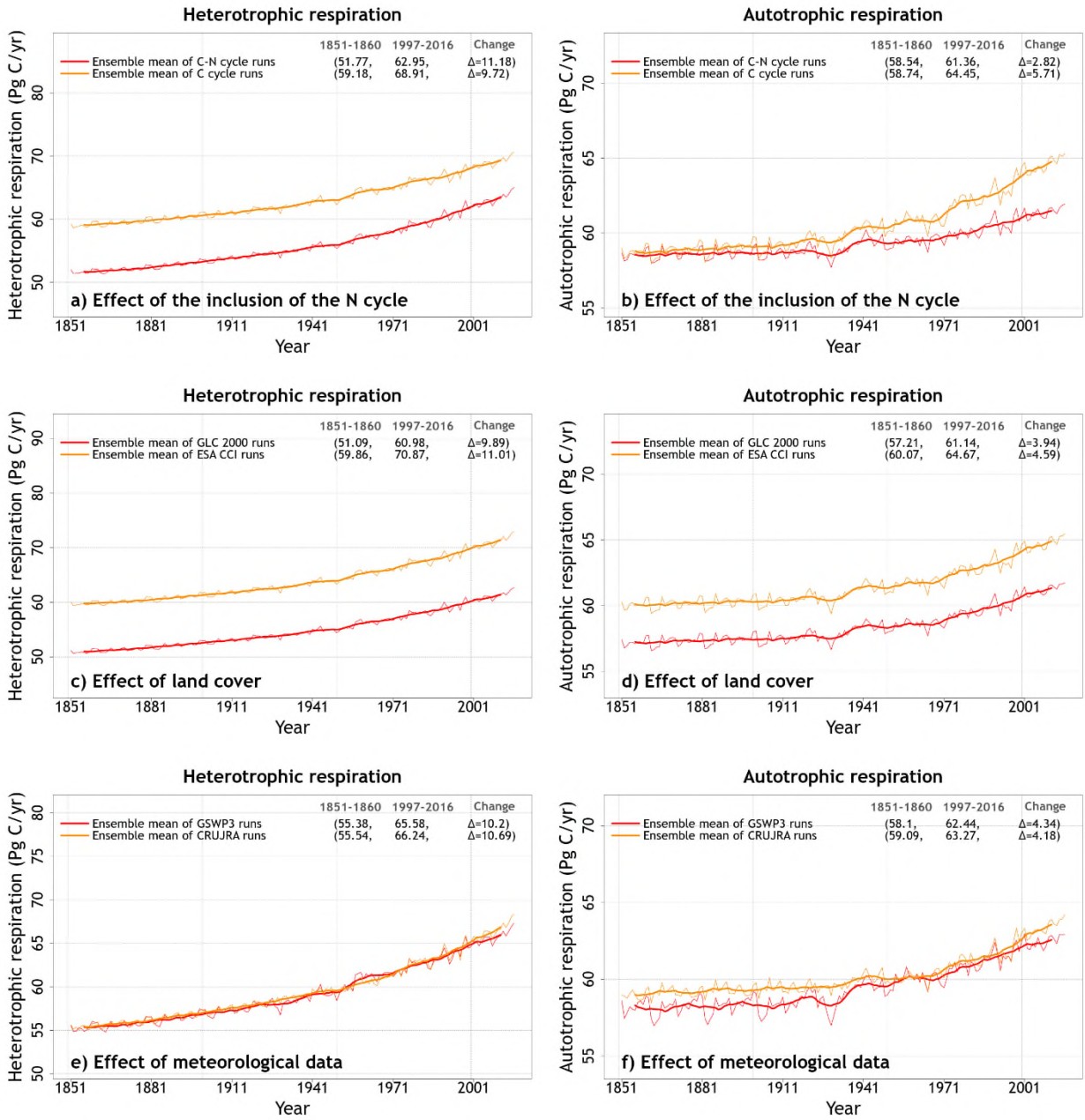

Figure A8: Time series of global heterotrophic and autotrophic respiration (over all land area
excluding Greenland and Antarctica) averaged over the four ensemble members each that are
driven with and without an interactive N cycle (panels a, b), driven with the GLC 2000 and ESA
CCI based land cover (panels c, d), and driven the with GSWP3 and CRU-JRA meteorological
data (panels e, f). The thin lines show the individual years and the thick lines show their 11-year
moving average. Model values averaged over the pre-industrial (1851-1860) and present-day
(1997-2016) time periods, and their difference, are also shown.


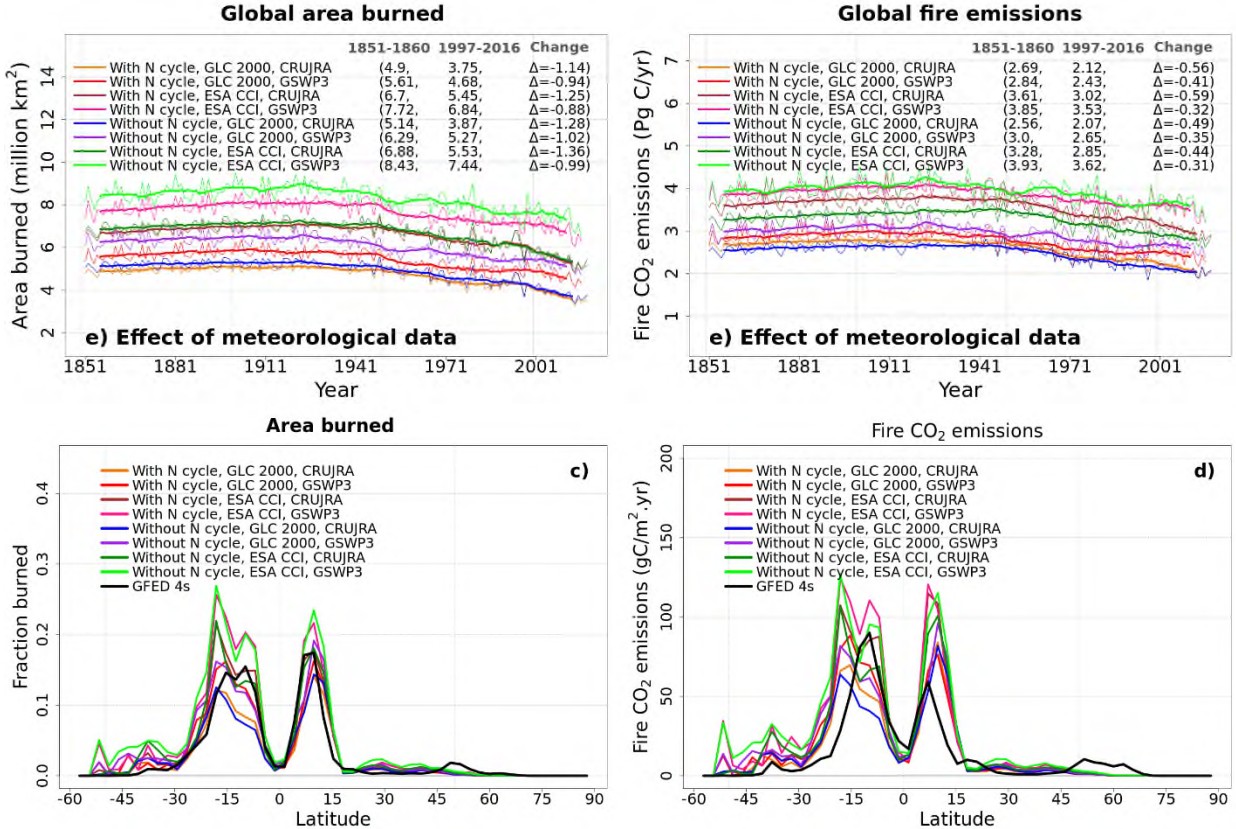



Figure A9: Time series of simulated global annual area burned (a) and fire $CO_2$ emissions (b)
from the eight simulations summarized in Table 1. Panels (c) and (d) show the zonally-averaged
area burned and fire $CO_2$ emissions from the eight simulations averaged over the 1997-2016
period. The thin lines for the time series show the individual years and the thick lines show their
11-year moving average. Model values averaged over the pre-industrial (1851-1860) and
present-day (1997-2016) time periods, and their difference, are also shown for panels (a) and
(b).


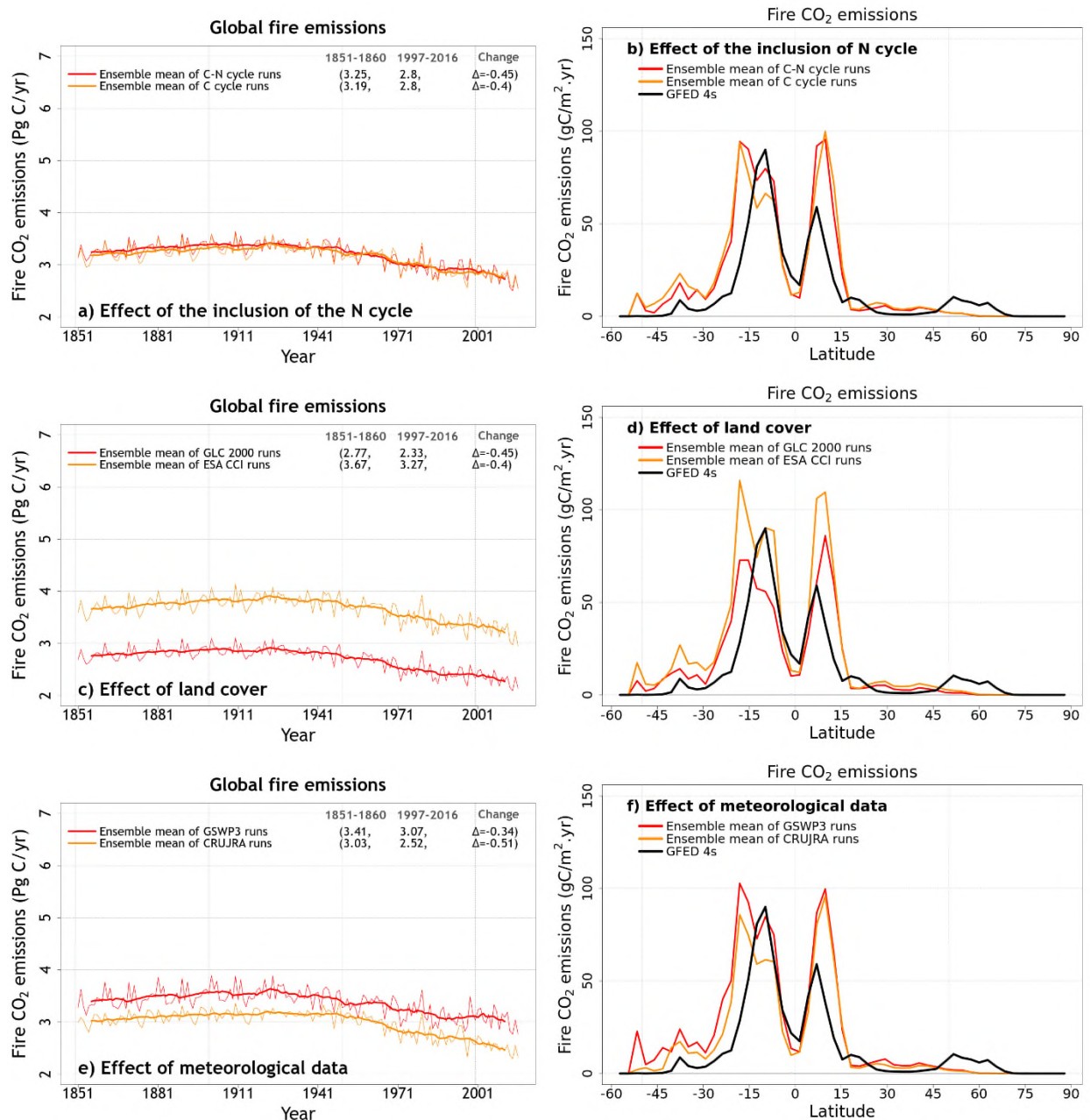


Figure A10: Time series of global fire $CO_2$ emissions (over all land area excluding Greenland and
Antarctica) (panels a, c, and e) and their zonally-averaged values (panels b, d, and f) averaged
over the four ensemble members each that are driven with and without an interactive N cycle
(panels a, b), driven with the GLC 2000 and ESA CCI based land cover (panels c, d), and driven
with GSWP3 and CRU-JRA meteorological data (panels e,f). The thin lines for the time series
show the individual years and the thick lines show their 11-year moving average in panels (a),
(c), and (e). Model values averaged over the pre-industrial (1851-1860) and present-day (1997-
2016) time periods, and their difference, are also shown for panels (a), (c), and (e).

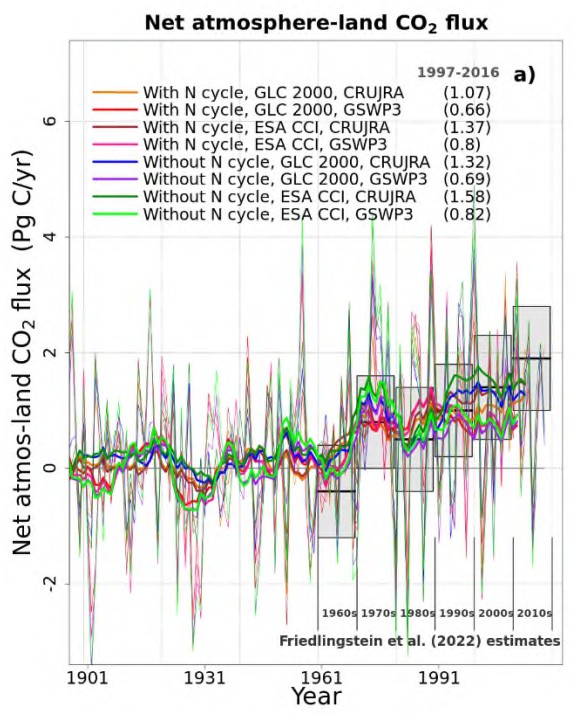
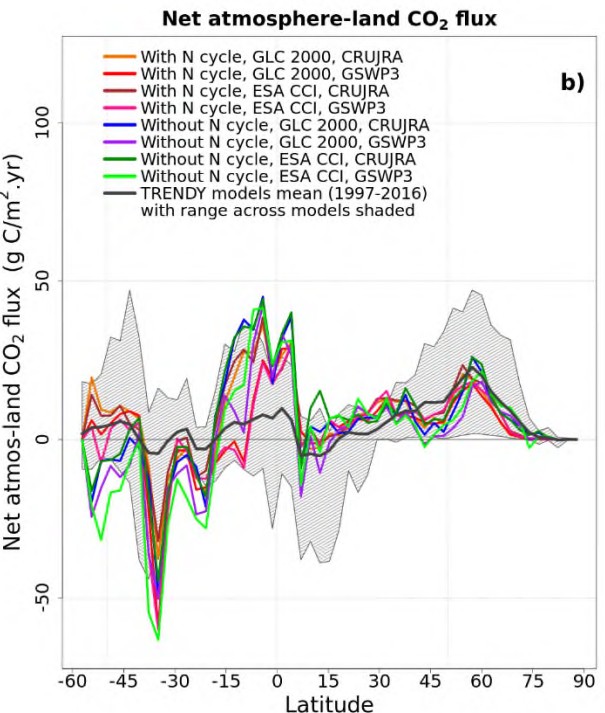



Figure A11: Time series of simulated global annual net atmosphere-land $CO_2$ flux (a) and its
zonally-averaged values from the eight simulations summarized in Table 1 averaged over the
1997-2016 period. In panel (a) simulated annual net atmosphere-land $CO_2$ flux values are
compared to the estimates from the Global Carbon Project (Friedlingstein et al., 2022). The thin
lines for the time series in panel (a) show the individual years and the thick lines show their 11-
year moving average. In panel (b) the simulated zonally-averaged values are compared to the
range from 11 models that contributed to the TRENDY 2020 intercomparison and averaged
over the 1997-2016 period.







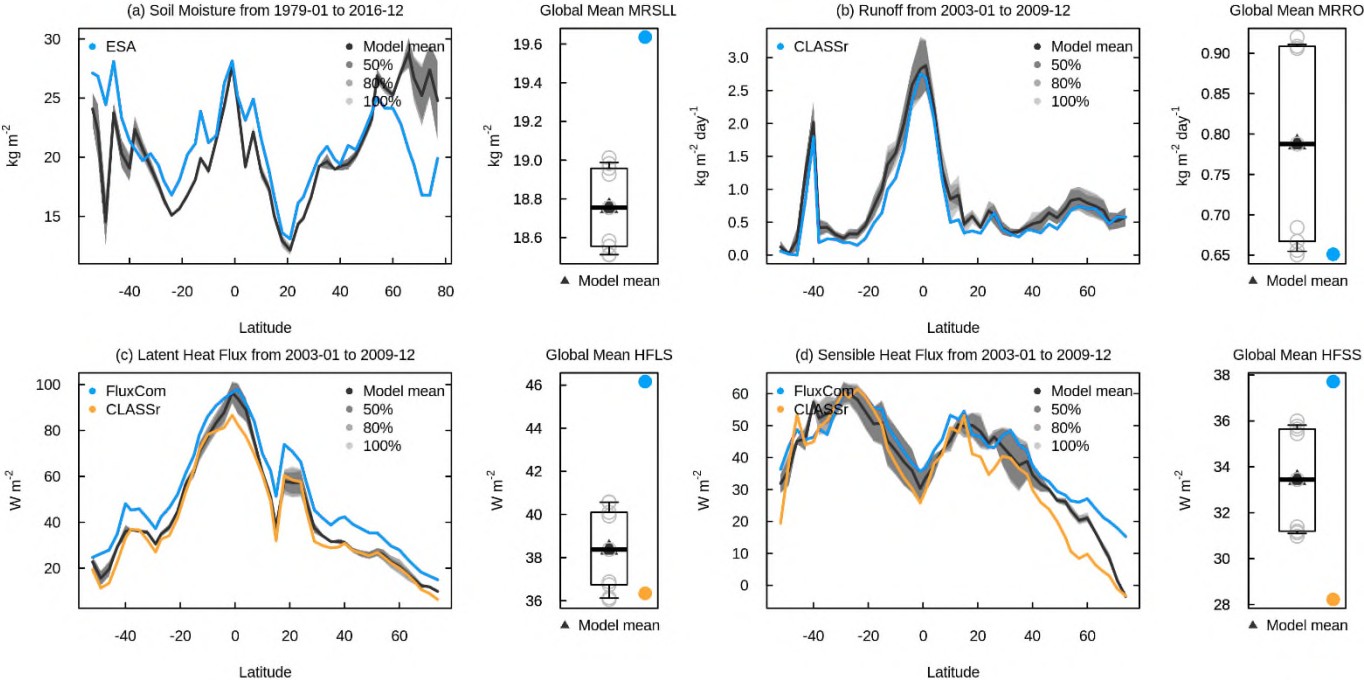



Figure A12: Zonally-averaged values of soil moisture (a), runoff (b), latent heat flux (c), and sensible heat flux (d) from the eight simulations summarized in Table 1. The model results are shown as their mean (black) and the spread across the eight simulations indicated by 50%, 80%, and 100% ranges in different shades of grey. The observation-based estimates used in AMBER to calculate scores are shown in coloured lines.

1526

1527

1528

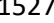

Figure A13: Zonally-averaged values of surface albedo (a), snow water equivalent (b), net surface radiation (c), net longwave radiation (d), and net shortwave radiation (e) from the eight simulations summarized in Table 1. The model results are shown as their mean (black) and the spread across the eight simulations indicated by 50%, 80%, and 100% ranges in different shades of grey. The observation-based estimates used in AMBER to calculate scores are shown in coloured lines.

1536

1537