# Peer review of "Towards an ensemble-based evaluation of land surface models in light of uncertain forcings and observations"

_EGUsphere, 2022_

## Author Response (AR1)

Revisions for EGUSPHERE-2022-641

Towards an ensemble-based evaluation of land surface models in light of uncertain forcings and observations

8 November 2022

Dear Associate Editor,

Thank you for giving us the opportunity to revise our manuscript following the reviewers' comments. We have taken all of the reviewers' comments into consideration while revising our manuscript as detailed in our response to the reviewers' comments in the Discussion section of our manuscript available online.
https://egusphere.copernicus.org/preprints/2022/egusphere-2022-641/

In particular, all model results figures have been modified to address the reviewers' concerns. Following their advice, we have moved the figures that showed results from all simulations to the appendix and moved the figures that showed ensemble-averaged results to the main text. We have also added the description of our model benchmarking approach to the appendix.

We have modified text at multiple places to provide additional information and clarifications to satisfy the reviewers' concerns.

Reviewer #1 raised concern that our approach does not take into account the uncertainty associated with the implementation of land use change. We have made this explicitly clear in the abstract, as well as in the main text and the Conclusions section.

Many thanks,
Vivek.

---

## Author Response (AR2)

We thank both reviewers for going through our manuscript again with a fine-toothed comb. Their comments are most certainly appreciated and have helped improve our manuscript tremendously. Manuscript reviewing is a volunteer service and we gratefully acknowledge the reviewers' time. We also apologize for not referring to page and line numbers in our response last time which we acknowledge caused reviewers to spend more time.

In the following text, reviewers' comments are highlighted in italics and our response is shown in bold. The line numbers are also mentioned and correspond to the revised manuscript with track changes.

**Reviewer # 1**

The authors have addressed my concerns with this revision, and in doing so provided some interesting additional analysis to their results.

I do have one 'technical correction' question related to how the phase score is described in Appendix A3. Equation A11 appears to have theta with units of the variable (as the max of the variable as represented by its climatological mean cycle), as opposed to units of days, as suggested by equation A12. If I am understanding this correctly, the notation for A11 should indicate a function that returns the difference in the day of year at which the max value of each of the two averaged data sets occurs.

Thank you for identifying this. In equation A11, the unit of theta is month and this has now been corrected. We have modified equations A11 and A12 on page 64.

**Reviewer #2**

The authors have done a good job responding to most of the reviewer comments, though it appears that not all the changes mentioned in the "Reply to Referee" comments have made it into the revised manuscript. This led to some difficulty assessing the revised manuscript in the context of the "Reply to Referee" comments, which were published before the manuscript was revised. I think what is missing is a clear link from reviewer comments to specific parts of the manuscript that were revised to address those comments (e.g., with new line numbers). The "Author's Response" document associated with the revised manuscript does not specifically mention what changes were made to address the reviewer comments, and instead points to the older "Reply to Referee" comments in the online discussion. I noted some discrepancies in the specific comments below, and hopefully the authors can clarify with future revisions.

Specific comments:

*Line 18: Please clarify "relative to their mean" – ensemble means? Annual means?*

The sentence in the abstract has now been modified as follows

... show the largest spread across the eight simulations relative to their GLOBAL ENSEMBLE mean VALUES (line 18-19 in the abstract).

Lines 182-194: I appreciate the clarification here regarding PFT-specific information for physical vs. biogeochemical processes. However, I'm not entirely convinced by this line: "For example, the interception of rain and snow by canopy leaves (that is typically modelled as a function of LAI and a PFTdependent parameter that accounts for leaf orientation and shape) does not depend on the underlying evergreen or deciduous nature of the leaf phenology."

True for phenology, but I would think that evergreen vs. deciduous would matter for interception via the PFT-dependent parameter accounting for leaf orientation and shape (e.g., needle vs. circular shapes). I'll also note that Reviewer #1 had a similar comment about how leaf/canopy shapes, orientations, and colors impact not only interception but also radiation, which can impact sensible heat flux. I think this section could be clarified further, perhaps by focusing in specific terms on what the CLASSIC model is doing (versus phrasing about LSMs in general).

**We have modified this paragraph in Section 2.1.2 (lines 180 to 190) to make it explicitly clear that this reasoning applies specifically to CLASSIC and not land surface models in general.**

Lines 255-256: It would be helpful to specify here what "a snapshot" means – is it matching the time period of the remotely sensed land cover product (e.g., 2000)? How do you go forward in time to create the historical land cover from 2000 to 2018? A brief line describing that process here would be helpful to round out this description.

We have added additional sentences in Section 3.1 (lines 263 and 264) to make it clear that in the context of the GLC 2000 data the snapshot in time indeed refers to year 2000. The text in this section does mention how we go forward in time to create land cover from 2000 to 2018 (lines 274 to 275). We have extended this sentence to make this more clear.

*Lines 329-338: These references are a welcome addition, though they may fit better in the introduction.*

We believe these references are more appropriate here since our process of land cover generated is described here and provides context.

Table 1: The simulation labels (A, B, C, etc.) seem useful but I couldn't determine whether they were actually used in the manuscript. It would make sense to use them in the legends of some of the appendix figures (e.g., Figure A3 and similar).

Thanks for catching this. The legend in the appendix figures makes it clear what forcings a given run uses so the simulation labels are somewhat redundant. We have changed the simulation labels from A, B, C ... to 1, 2, 3 ... in Table 1 so that they look like a list rather than something intended to be referred to later.

Line 410: In the "Reply to Referee" document in response to a question about different end dates for simulations with different meteorological forcings, it is mentioned that the authors "will make the time period the same for a consistent comparison". However, this line still mentions different end years for the

different forcing data: 2018 vs. 2016. The timeseries figures do appear to end at the same year, with a consistent 1997-2016 average calculated.

The simulations do end at different years since the two meteorological data have different lengths but we have now kept the comparison period 1997-2016 the same.

*Line 433: 19 variables are mentioned but only 18 are listed - is ecosystem respiration missing from the list?*

The variables used for the AMBER scores are most easily identified in Figure 10 which contains 16 variables. The three variables missing from here are soil moisture, ecosystem respiration, and fire CO2 emissions because for these variables only one observation-based reference data set is available so a benchmark score can't be calculated. This is now clarified in the revised manuscript (Section 4.3, lines 782 to 796).

Table 2: I'm not seeing the changes to Table 2 that were mentioned in the "Reply to Referee" document: "We will group by variable and reorganize Table 2 according to whether the data are globally gridded and/or in situ, or make note of this in an additional column". I'm also not seeing a table entry for fire CO2 emissions (as mentioned in line 436).

After writing the reply to the referees' first round of comments we realized that Table 2 has always indicated which variables are globally gridded and which are not. See the screenshot below. Thank you for catching fire  $CO_2$  emissions which we have now included Table 2.

| Globally gridded variable(s)           | Source         | Approach used                                                 | Reference                            |
|----------------------------------------|----------------|---------------------------------------------------------------|--------------------------------------|
| Leaf area index                        | AVHRR          | Artificial neural network                                     | Claverie et al. (2016)               |
| 61-a la 1                              | CANAC          | A                                                             | A                                    |
|                                        |                |                                                               | 1                                    |
| In situ variable(s)                    | Source         | Approach used (number of sites)                               | Reference                            |
| In situ variable(s)
Leaf area index | Source
CEOS | Approach used (number of
sites)
Transfer function (141) | Reference
Garrigues et al. (2008) |

Lines 463-466: Suggest adding a reference to Table 3 here since that is where the cv results are reported.

**Thank you. We have added the reference to Table 3 here. Line 488 in the revised manuscript.**

Line 615: Heterotrophic respiration does not seem affected by meteorological data, but it's interesting that autotrophic respiration is a little affected (Fig A8 panel f), almost on a similar scale to the impact of an interactive N cycle (Fig A8 panel b). It could be worth noting that point as well as the fact that the impacts vary over time which is somewhat different than the other carbon variables.

**We have added a sentence towards the end of section 4.2.1 (lines 665-667), just before section 4.2.2, to note the difference in the effect of the meteorological data on heterotrophic and autotrophic respiratory fluxes.**

*Lines 626-629: Similar to my comment in the first round about the sentence at the end of section 4.1, I think this sentence could also be removed for conciseness since Table 3 covers the summary of cv values and drivers.*

**This last paragraph of section 4.2.1 has now been deleted.**

*Lines 665-668: This explanation is helpful to understand what is shown in Table 3 vs. Table 2. It might be worth listing here the other variables that are not included in Table 3 (net radiation, net ecosystem exchange) for completeness.*

Net radiation and net ecosystem exchange are now mentioned in Section 4.2.3 (Coefficient of variation summary) (lines 711-715).

*Lines 669-671: These sentences seem a little out of place, and are perhaps better suited to the conclusions section.*

Line 843: In the "Reply to Referee" document in response to a comment about linking the conclusions to specific results, it is mentioned that the authors "will cover model tuning early on in the manuscript and link it to the second paragraph in the conclusions". However, I was unable to find where this topic had been added to the revised manuscript. There is Line 670-671 that seems better suited to this paragraph in the conclusions (as mentioned above), but nothing in the introduction.

In regards to both the above two comments, we have added a new section 4.2.4 (lines 718-725) that explicitly talks about model tuning. We believe this is an ideal place for this short section because at this point in the manuscript it is clear to a reader how different meteorological data sets, land cover representations, and model versions affect simulated model quantities in different ways.

Please also note that lines 124-126 in the Introduction section do briefly raise the issue of model tuning.

*Lines 880-884: It would be helpful to briefly summarize here the reasons for the effects of different forcing datasets (e.g., I believe precipitation/wind differences are mentioned in the results).*

We have added few more sentences around these lines. We believe that high precipitation intensities and warmer northern tropical region in the GSWP3, compared to the CRU-JRA,

meteorological data are the likely cause for these differences as explained in the revised text (lines 964-974).

**Technical corrections:**

*Line 362: Suggest changing the precipitation units to mm/month to match the values in Figure A1.*

**Done**

*Line 379: The phrase "time-invariant monthly lightning" is unclear, perhaps "monthly climatological lightning" or "prescribed monthly lightning"?*

**Done**

Line 414: Suggest specifying "128 x 64 grid cells".

**Done**

*Line 674: Please check the years here – Figure A11 caption says the zonal means are averaged over 1997-2016.*

**Thank you for catching this.**

NBP Figures (A11 and 9): It would help to decrease the y-axis scales of the NBP figures so that the differences can be shown more clearly. The legends can be moved outside of the figure area if needed in order to decrease the y-axis upper limit. This is somewhat of an issue throughout all the figures in the manuscript, but especially apparent here where it's difficult to distinguish the NBP variations.

Since NBP is an important variable, we have taken reviewer #2's suggestion. However, rather than moving the legend outside (which apparently is not straightforward in R) we have lengthened the figure so that the variations in NBP are more clearly visible in Figure A11. Moving the legend out of the plotting area would have meant a longer figure as well. We have also changed the y-axis maximum in Figure 9 from 8 to 6 Pg C/yr so that the plots in Figure 9 a), c), and e) are more readable.

Zonal mean Figures: Not all the zonal mean figures specify over which time period the zonal mean averaging was computed (e.g., Figures 5, 6, 7, 8, 9, A10). At first I thought this was the entire time period, but some of the appendix figures (e.g., Figures A6, A7, A9, A11) specify that the zonal mean values are over the present day period. Some clarification in the figure captions is needed.

Yes, zonal averages are calculated over the common 1997-2016 period. We now note this in the figure captions for Figures 5, 6, 7, 8, 9, A10. Thanks for catching this.

Line 728: Suggest moving this line ("The range in model scores...") up to Line 723 before the additional text describing the whiskers.

Done.

*Line 833: Something is off grammatically here, maybe add "and how the response is dependent..."*

Done.